# Loss of miR-210 leads to progressive retinal degeneration in *Drosophila melanogaster*

Carina M Weigelt[1], Oliver Hahn[1], Katharina Arlt[1], Matthias Gruhn[2], Annika J Jahn[1], Jacqueline Eßer[1], Jennifer A Werner[1], Corinna Klein[3], Ansgar Büschges[2], Sebastian Grönke[1], Linda Partridge[1,4]

**miRNAs are small, non-coding RNAs that regulate gene expression post-transcriptionally. We used small RNA sequencing to identify tissue-specific miRNAs in the adult brain, thorax, gut, and fat body of *Drosophila melanogaster*. One of the most brain-specific miRNAs that we identified was miR-210, an evolutionarily highly conserved miRNA implicated in the regulation of hypoxia in mammals. In *Drosophila*, we show that miR-210 is specifically expressed in sensory organs, including photoreceptors. miR-210 knockout mutants are not sensitive toward hypoxia but show progressive degradation of photoreceptor cells, accompanied by decreased photoreceptor potential, demonstrating an important function of miR-210 in photoreceptor maintenance and survival.**

## Introduction

miRNAs are small, non-coding RNAs of approximately 22 nucleotides that regulate gene expression post-transcriptionally. miRNAs bind their targets by complementary seed matches in the 3'-UTR of mRNAs. Regulation of mRNA expression by miRNAs occurs through two different post-transcriptional mechanisms: mRNA cleavage or translational repression (Bartel, 2009). miRNAs are often expressed in a specific cell type associated with their function. For example, miR-124 is brain specifically expressed and plays a role in brain function in diverse species ranging from *Caenorhabditis elegans* and *Drosophila* to mammals (Kapsimali et al, 2007; Clark et al, 2010; Weng & Cohen, 2012). Hence, identification of tissue-specific miRNAs is crucial to understand their function. Previous studies have analysed miRNA expression pattern in whole bodies, heads, and ovaries/testes of *Drosophila* (Fagegaltier et al, 2014) and during *Drosophila* development (Aboobaker et al, 2005). More recently, FlyAtlas2 included also miRNA expression in certain tissues (Leader et al, 2018), but little is known about distribution of miRNAs in metabolically important adult *Drosophila* tissues such as brain, muscles, fat body, and gut.

In this study, we used next-generation sequencing of small RNAs to identify tissue-specific miRNAs in adult brain, thorax, gut, and fat body tissues of 10 d old, wild-type *Drosophila* flies. We identified many brain-specific miRNAs, including the highly evolutionarily conserved miR-210. miR-210 has been intensively studied in the context of the response to hypoxia in mammalian cell culture (Camps et al, 2008; Fasanaro et al, 2008; Giannakakis et al, 2008; Pulkkinen et al, 2008; Chan et al, 2009; Huang et al, 2009). Furthermore, several mouse studies have verified that miR-210 is also up-regulated in hypoxic conditions in vivo in models for ischemia or pulmonary hypertension (Pulkkinen et al, 2008; Zaccagnini et al, 2014; White et al, 2015). Recently, several studies linked miR-210 to the circadian clock in *Drosophila*, as it is up-regulated in *cyc[01]* mutants, which have an impaired circadian clock (Yang et al, 2008). Furthermore, overexpression of miR-210 affected circadian locomotor activity in *Drosophila* (Cusumano et al, 2018; You et al, 2018).

We have found that miR-210 is specifically expressed in photoreceptors, ocelli, and the antennal lobes. Loss of miR-210 led to progressive loss of photoreceptor integrity, accompanied by reduced photoreceptor function as measured by electroretinography. Furthermore, we used RNA sequencing to identify putative miR-210 target genes. Altogether, we have produced an expression atlas for miRNAs in adult *Drosophila* tissues, and we describe a novel function for miR-210 in vivo in photoreceptor maintenance.

## Results

### Identification of tissue-specific miRNAs by small RNA sequencing

To generate a *Drosophila* miRNA expression atlas for adult tissues, we used next-generation sequencing on dissected brain, thorax, gut, and fat body of 10-d-old, female wild-type flies (n = 3) (Supplemental Data 1). We evaluated tissue specificity of single miRNAs by a tissue specificity score, similar to a previous approach to identify tissue-specific miRNAs in mammals (Landgraf et al, 2007). Of the total 184 detected miRNAs, 75 showed a highly tissue-specific

[1]Max Planck Institute for Biology of Ageing, Cologne, Germany   [2]Department for Animal Physiology, Biocenter Cologne, Institute of Zoology, Cologne, Germany   [3]Cluster of Excellence—Cellular Stress Responses in Aging-Associated Diseases Research Centre, University of Cologne, Cologne, Germany   [4]Institute of Healthy Ageing, Genetics, Evolution and Environment, University College London, London, UK

Correspondence: sgroenke@age.mpg.de; Partridge@age.mpg.de

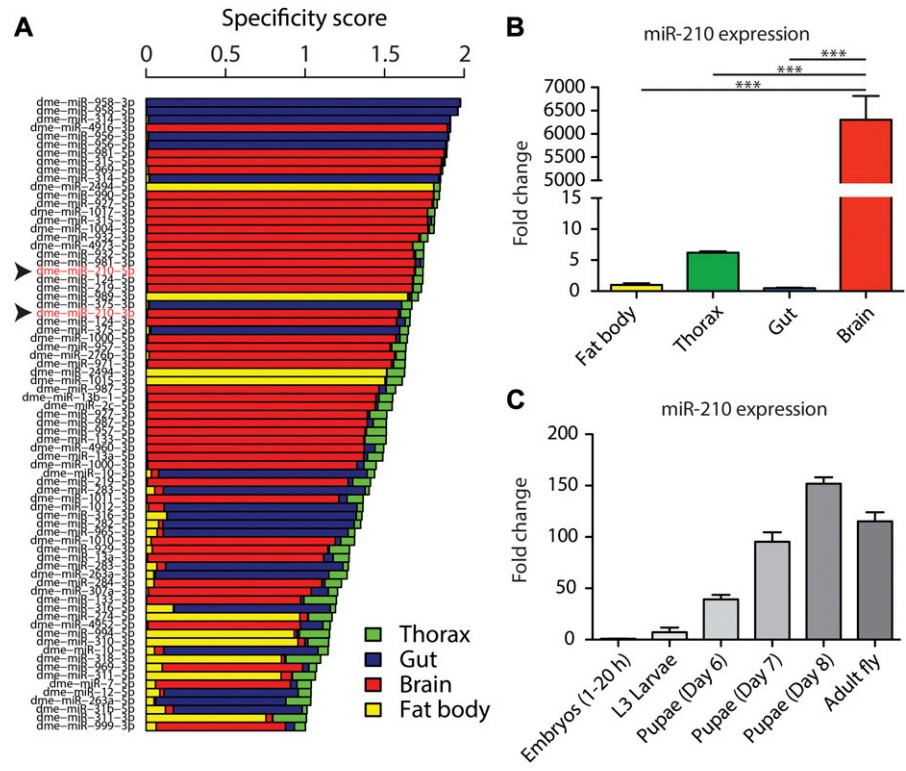

**Figure 1. Tissue-specific expression atlas of miRNAs in adult *Drosophila*.**
**(A)** Small RNA sequencing of brain, thorax, gut, and fat body tissue of 10-d-old, female, wild-type flies revealed tissue-specific miRNAs (*n* = 3). Top 50 tissue-specific miRNAs are shown (red = brain, green = thorax, blue = gut, and yellow = fat body). **(B)** qRT-PCR verified that miR-210 is highly brain specifically expressed in 10-d-old, female, wild-type flies (***P < 0.001; one-way ANOVA, *n* = 3). **(C)** miR-210 was lowly expressed during embryonic and larval development and increased in expression at late pupal stages and in adult flies. Up to 20-h-old embryos and wandering L3 larvae were used. We used whole animals for this experiment (*n* = 3).

expression pattern (Fig 1A), with 44 brain-specific, 21 gut-specific, and 10 fat body–specific miRNAs. Most miRNAs with tissue-specific expression were preferentially expressed in the brain. Our RNA sequencing approach verified the expression pattern of several well-studied miRNAs, for example, miR-124, which is highly brain-specific from worms to mammals and plays an important role in neuronal development and function (Kapsimali et al, 2007; Clark et al, 2010; Weng & Cohen, 2012). Moreover, our analysis also revealed tissue-specific expression of several less-studied miRNAs, indicating a potential function for them. For example, miR-958 was the most gut-specific miRNA detected in our study, and initial studies have linked miR-958 to the *Drosophila* innate immune system (Li et al, 2017). Our results suggest that the gut-specific miR-958 might contribute to the gut-specific responses to bacterial infection. Another interesting gut-specific miRNA is miR-314, which has been previously studied in the midgut upon exposure to xenobiotics (Chandra et al, 2015), verifying that miR-314 indeed plays an important function in the gut. No miRNA reached the tissue-specificity threshold in the thorax, but we identified several miRNAs that were at least enriched in the thorax, including the well-studied miR-1, which is specifically expressed in muscle from worms to humans (Kwon et al, 2005; Sokol & Ambros, 2005; Zhao et al, 2005; Chen et al, 2006; Simon et al, 2008).

Among the tissue-specific miRNAs, we identified the highly expressed and probably active dme-miR-210-3p and the lower expressed dme-miR-210-5p*, which were both specifically expressed in the brain. MiR-210 is an evolutionarily highly conserved miRNA that in mammals has been implicated in response to hypoxic conditions. However, a function for miR-210 in the brain is currently unknown. Thus, we decided to further investigate the role of miR-210

in the fly brain. We first verified the brain-specific expression pattern of miR-210 by miRNA quantitative real-time PCR (qRT-PCR) (***P < 0.001 brain versus fat body/thorax/gut, n = 3) (Fig 1B) and found that miR-210 was only weakly expressed during development but was activated during late pupal stages and maintained active in adult flies (n = 3) (Fig 1C). The brain-specific expression of miR-210 in adult flies suggests that it might have a specific role in the maintenance of adult brain function.

### miR-210 function is not essential for survival under hypoxic conditions in flies

In mammals, miR-210 expression is up-regulated under hypoxia by the transcription factor HIF-1α (Camps et al, 2008; Fasanaro et al, 2008; Pulkkinen et al, 2008). The hypoxia pathway, including HIF-1α, is highly conserved between flies and mammals (Lavista-Llanos et al, 2002), but it is currently unclear if miR-210 is also involved in the response to hypoxia in flies. Therefore, we used qRT-PCR to measure whether miR-210 is also induced by hypoxia in adult flies. However, miR-210 expression was not induced when flies were exposed to 6 h of 2.5% O$_2$ (Fig S1A), which was sufficient to activate the expression of the well-known HIF-1α target Scylla (**P < 0.01, n = 3) (Reiling & Hafen, 2004). To further study the potential role of miR-210 in hypoxia in *Drosophila*, we used CRISPR/Cas9-mediated genomic engineering to generate an miR-210 null mutant, termed *miR-210Δ*. Lack of miR-210 expression in the miR-210 null mutant was verified by qRT-PCR (****P < 0.0001, n = 3) (Fig S1B). *miR-210Δ* mutant larvae or adult flies behaved as wDahomey (wDah) wild-type flies under hypoxic conditions, in contrast to the positive control, heterozygous Sima (HIF-1α) mutant larvae and adult flies

that showed decreased survival (\*\*\*$P < 0.001$) (Fig S1C and D). This result suggests that miR-210 is not an essential mediator of the hypoxia response in flies. Hypoxia is a potent regulator of target of rapamycin (TOR) signalling (Arsham et al, 2003). Therefore, we also investigated TOR-related phenotypes in the *miR-210Δ* mutant flies. In line with the hypoxia experiments, we found no striking differences between *miR-210Δ* and control flies in lifespan (n = 200, females: n.s., and males: \*\*$P < 0.01$) (Fig S1E). This result is in contrast to a recent study, which reports decreased lifespan of *miR-210* mutant males (Chen et al, 2014). Differences in genetic background or lack of adjusting the genetic background in the study by Chen et al (2014) might explain this discrepancy. In line with the lifespan results, starvation stress resistance (n = 100), body weight (n~50, \*$P < 0.05$), and egg laying (n = 100) were also not affected by lack of *miR-210* (Fig S1F–H), suggesting that *miR-210Δ* does not influence TOR signalling systemically. In summary, in contrast to mammalian cell culture, where miR-210 is up-regulated under hypoxia (Camps et al, 2008; Fasanaro et al, 2008; Pulkkinen et al, 2008), we did not detect induction of miR-210 in vivo when exposing flies to hypoxic conditions, and resistance to hypoxia was unaffected by its absence. Consistently, a screen for hypoxia-regulated miRNAs in *Drosophila* did not identify miR-210 (De Lella Ezcurra et al, 2016). Thus, the function of miR-210 in response to hypoxia might not be conserved between flies and mammals. However, we currently cannot exclude that miR-210 is only regulated by hypoxia in a subset of cells and, therefore, might cause only local effects.

## Loss of miR-210 leads to retinal degeneration

To dissect the function of miR-210 in the brain, we used CRISPR/Cas9 to generate an *miR-210Δ GFP* reporter line (*miR-210Δ GFP*) to study in which cells of the brain miR-210 is expressed. Interestingly, miR-210Δ GFP expression was highly specific to the fly compound eye, the ocelli, and the antennal lobes (Fig 2A), which are important for sensing light and olfactory cues. By co-immunostaining miR-210Δ GFP with the photoreceptor marker chaoptin (Fujita et al, 1982), we demonstrated that miR-210 is expressed in photoreceptor cells projecting into the lamina and medulla of the fly optic lobes (Fig 2A and B). By whole-mount retina staining and cryosections of miR-210Δ GFP heads, we further demonstrated that miR-210 is also expressed in the fly retina, including photoreceptors and potentially pigment cells (Fig 2C and D). During preparation of this manuscript, another study showed a similar expression pattern for miR-210 in the fly brain (Cusumano et al, 2018), validating our findings. Thus, we showed that miR-210 is specifically expressed in the sensory organs of the fly, including the retina.

The function of the fly eye is well characterized, and the pathways involved in phototransduction are partially conserved between flies and mammals (Xu et al, 1999; Montell, 2012; Sen et al, 2013). Given its specific expression in the fly eye, we wondered if the morphology and function of the fly eye is altered in *miR-210Δ* mutant flies. In fly retinas, R1-R6 and R7 can be visualized by chaoptin immunostaining. Throughout life, wild-type flies retained a very well-structured and organized pattern of R1-R6 and R7 in each ommatidium (Fig 3A). Retinas of very young (day 0) *miR-210Δ* mutants showed an apparently normal arrangement of R1-R6 and R7 photoreceptor cells, suggesting that miR-210 function is not

essential for photoreceptor differentiation. However, already at 10 d of age, *miR-210Δ* mutant photoreceptor cells showed altered arrangement and morphology, and at 42 d of age, individual photoreceptor cells could not be identified anymore (Fig 3A). To quantify the functional decline of photoreceptor cells in *miR-210Δ* mutants, we used electroretinography to measure the receptor potential in photoreceptor cells (n = 5-8). In line with immunostaining, wild-type flies showed a stable receptor potential that did not decline even late in life. By contrast, in *miR-210Δ* flies, the receptor potential decreased strongly with age, verifying that there was functional decline (\*\*\*\*$P < 0.0001$) (Fig 3B and C). Next, we used toluidine-stained semi-thin sections and transmission electron microscopy (TEM) to investigate the retinal degeneration in more detail. We used a closer time window (1 h, 2 d, 4 d, and 10 d) to obtain a better temporal resolution. In line with our previous immunostainings, *miR-210Δ* photoreceptor cells degenerated rapidly (Fig 3D). The higher resolution of TEM allowed us to detect differences in rhabdomere morphology of *miR-210Δ* mutants even in very young flies (1 h old, white arrows), which was not possible by immunostaining. Thus, lack of miR-210 might also mildly affect photoreceptor development. TEM also revealed the occurrence of vacuoles (arrowheads in Fig 3D), after the photoreceptor cells disappeared, which might implicate autophagic clearance of photoreceptor remnants. We further validated that the observed phenotype is caused by loss of *miR-210* by using an independently generated *miR-210* mutant line (*miR-210ΔSeed*), in which we deleted the functional seed sequence of *miR-210* via CRISPR/Cas9. In addition, we did a genetic complementation assay by crossing *miR-210Δ* mutants to a deficiency fly stock (Df(1)BSC352) encompassing the *miR-210* gene locus. Similar to *miR-210Δ* mutants, *miR-210ΔSeed* and *miR-210Δ*/Df(1)BSC352 mutant flies presented a strong retinal degeneration phenotype at 4 d of age (Fig S2A and B). In summary, we show that loss of *miR-210* leads to retinal degradation accompanied by a functional decline of photoreceptor neurons with age. The presence of R1-R6 and R7 photoreceptor cells and a wild-type–like receptor potential measured in freshly eclosed flies and the fast degradation of photoreceptor neurons within a few days suggest that miR-210 expression in the photoreceptors, lamina, and/or medulla is crucial for the maintenance and function of adult photoreceptor neurons.

To provide further evidence that the observed retinal degeneration phenotype was caused by loss of *miR-210* function, we performed rescue experiments by overexpression of miR-210 using the eye-specific GMR-Gal4 driver line. First, we showed that overexpression of miR-210 in the wild-type background per se did not lead to retinal degeneration or altered function of photoreceptors as measured by immunostainings and electroretinography (Fig S3A and B). However, we noted by TEM that overexpression of miR-210 in the eye led to several ommatidia that presented eight visible rhabdomers, which might be split rhabdomers (Fig S3C). Next, we showed by qRT-PCR that GMR-Gal4–mediated overexpression of miR-210 in the *miR-210Δ*–mutant background restored miR-210 expression to slightly higher levels than observed in wild-type flies (n = 3) (Fig 4A). Notably, overexpression of miR-210 rescued the *miR-210Δ*–dependent decline in photoreceptor potential. Receptor potential of *miR-210Δ*; *GMR-Gal4>UAS-miR-210* mutants was significantly increased compared with *miR-210Δ* controls (n = 7;

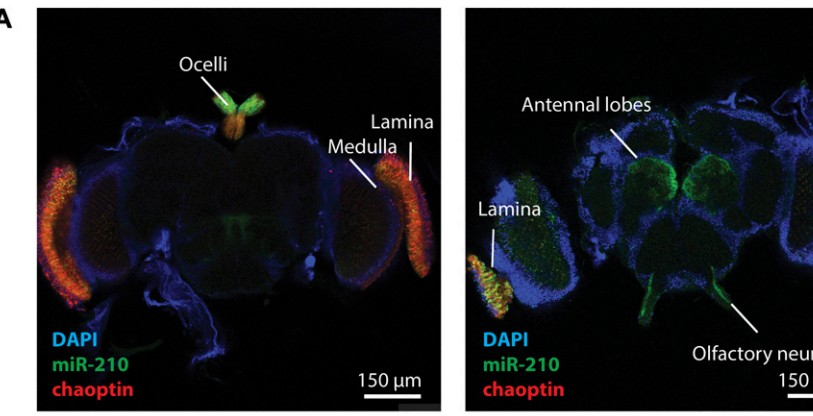

**Figure 2.  miR-210 is expressed in the fly eye.**
**(A, B)** Whole-mount immunostainings of miR-210Δ GFP reporter fly brains with anti-GFP antibody revealed a specific expression pattern of miR-210 (green) in the lamina and medulla of the optic lobes (left and right side of the brain), ocelli (top side of the brain), and antennal lobes (in the middle of the brain). miR-210 (green) expression patterns were overlapping with the photoreceptor marker chaoptin (red) in the medulla and lamina. **(C, D)** Whole-mount immunostainings and cryosections of miR-210Δ GFP reporter fly brains with anti-GFP antibody demonstrated that miR-210 is also expressed in the fly retina.

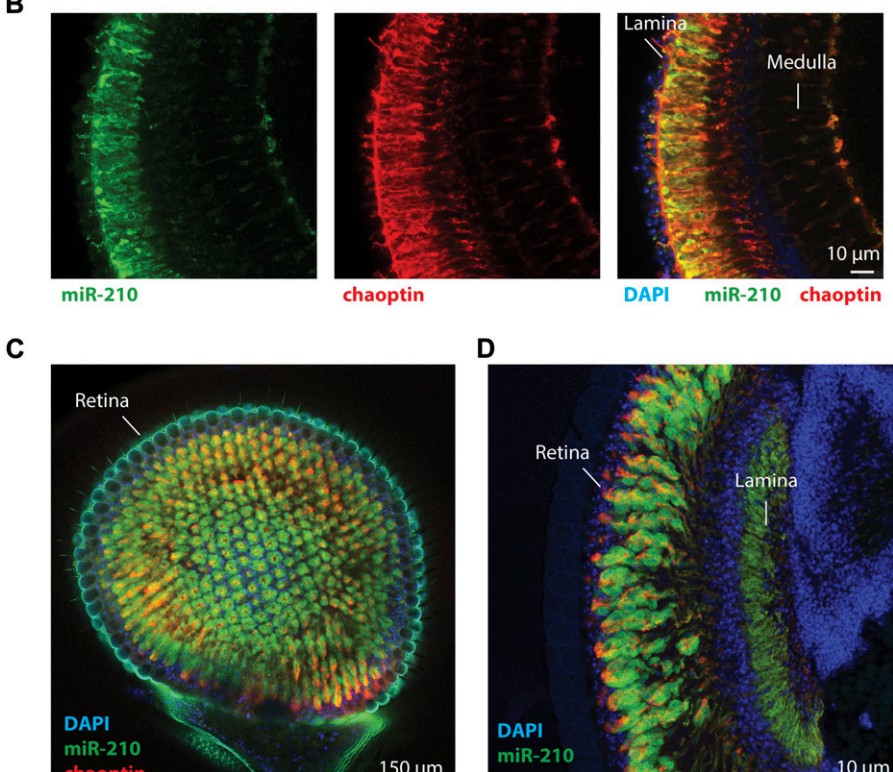

***P < 0.001) (Fig 4B). Surprisingly, *UAS-miR-210/+* control flies in the *miR-210Δ* background already had a significantly increased receptor potential compared with *GMR-Gal4/+* mutants in the *miR-210Δ* background (*P < 0.05), which might be explained by leaky expression from the UAS-promoter (Fig 4A). TEM of *miR-210Δ; GMR-Gal4>UAS-miR-210* flies also showed a partial rescue of the retinal degeneration phenotype and a reduction in the size and number of vacuoles (Fig 4C). Similar to the overexpression of miR-210 in the wild-type background, we also occasionally observed eight rhabdomeres upon overexpression of miR-210 in *miR-210Δ* mutants. In summary, we were able to at least partially rescue the retinal degeneration phenotype, demonstrating that the observed phenotype is indeed caused by a lack of *miR-210* function. That we only

obtained a partial rescue might be explained by differences between the expression of the GMR-Gal4 driver line and the endogenous expression pattern of miR-210 or by the mild phenotype of miR-210 overexpression alone.

## miR-210–mediated retinal degeneration is independent of light and apoptosis

miR-210 has been previously linked to the circadian clock (Yang et al, 2008, 2018; Cusumano et al, 2018), which is entrained by light. As miR-210 is expressed in the fly eye, we wondered if miR-210 expression changes rhythmically during the day and whether the retinal degeneration seen in *miR-210Δ* mutants might depend on

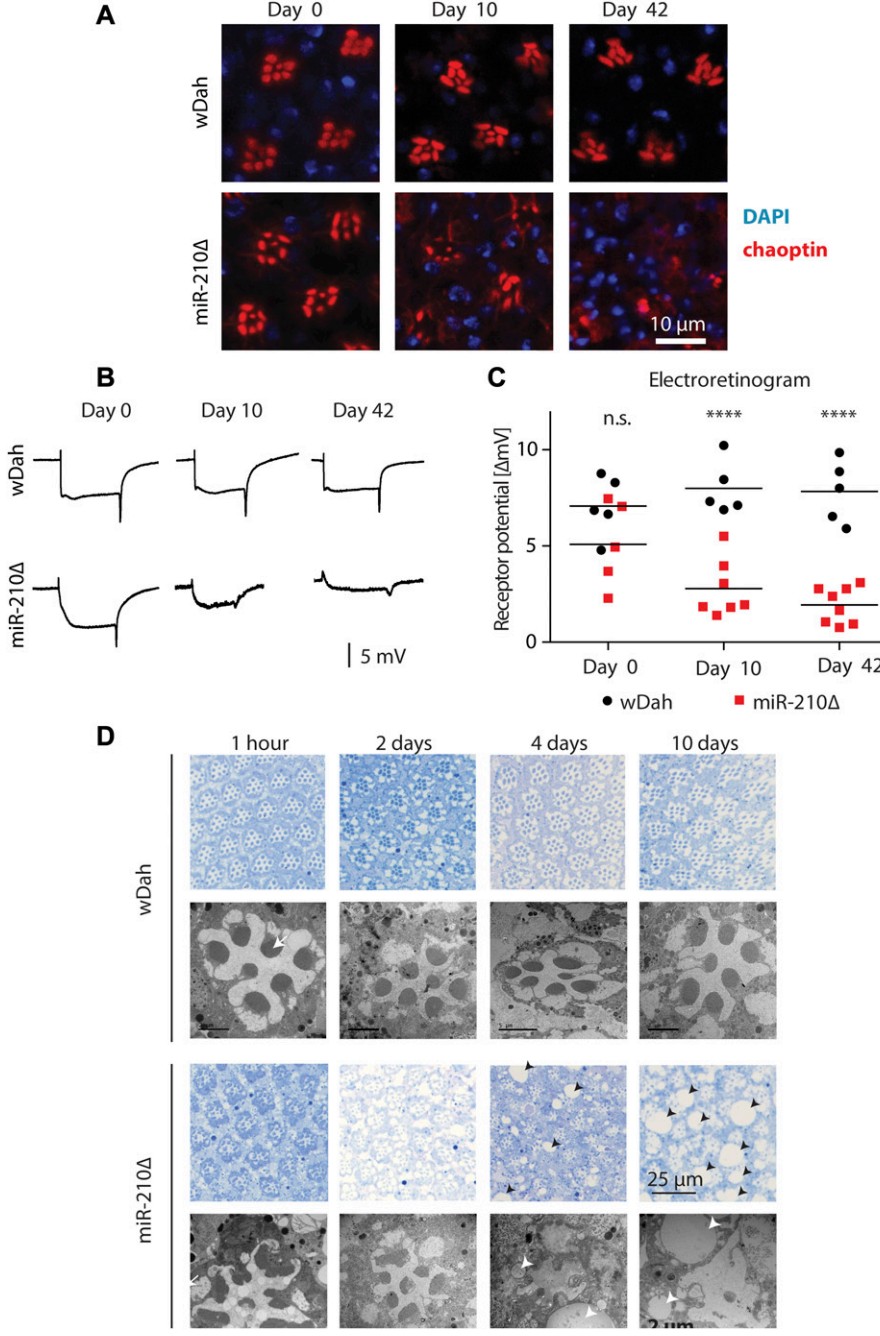

**Figure 3.  Loss of *miR-210* leads to progressive retinal degradation with age.**
**(A)** Immunostaining of retinas with chaoptin (red, marker for photoreceptors) showed loss of photoreceptor arrangement and integrity in *miR-210Δ* mutants, but not in wild-type controls, with age. **(B, C)** The receptor potential in *miR-210Δ* mutants, but not in controls, decreased with age as determined by electroretinography (age: n.s.; genotype: ****P < 0.0001; interaction: *P < 0.05; and two-way ANOVA, n = 5–8). **(D)** Toluidine-stained semi-thin sections (upper panel, blue) and TEM (lower panels) verified that photoreceptor cells progressively disappeared in *miR-210Δ* mutant, but not wild-type, eyes (1 h–10-d old flies). Notably, morphology of rhabdomeres (arrows) was already altered in 1-h-old *miR-210Δ* mutants. The number and size of vacuoles (arrowheads) increased with age in *miR-210Δ* mutants.

light, as has been shown for other mutations that cause retinal degeneration in the fly eye (Harris & Stark, 1977; Dolph et al, 1993; Kiselev et al, 2000; Johnson et al, 2002). However, miR-210 was not expressed in a circadian-dependent manner (n = 3) (Fig S4A), in line with a previous publication (Yang et al, 2008). In addition, miR-210 expression did not change in flies that were maintained for 48 h under constant light or constant darkness (n = 3) (Fig S4B), indicating that the expression of miR-210 is not regulated by light. Keeping *miR-210Δ* mutants under constant light or constant darkness during development and adulthood did not affect the retinal degeneration phenotype as compared with flies kept under 12 h/12 h light/dark conditions (Fig S4C), indicating that the mechanisms by which lack of *miR-210* affects retinal degeneration is not light dependent. Block of apoptosis rescues several retinal degeneration mutants (Kiselev et al, 2000; Johnson et al, 2002); however, block of apoptosis by overexpression of the antiapoptotic p35 protein did not rescue the receptor potential of *miR-210Δ* mutants (n = 5) (Fig S4D), suggesting that retinal degeneration caused by lack of *miR-210* function is not acting via p35-dependent induction of apoptosis. Thus, our results suggest that *miR-210*–mediated retinal

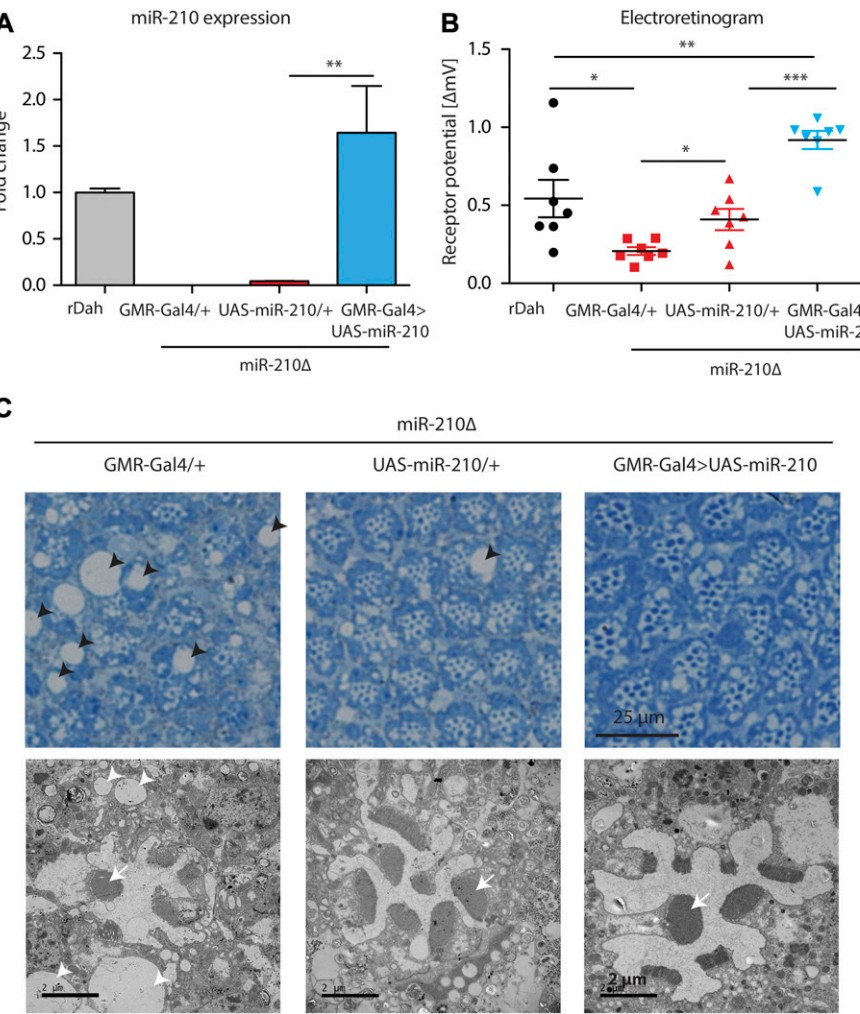

**Figure 4. Overexpression of miR-210 in *miR-210Δ* eyes partially rescued the photoreceptor degeneration.**

**(A)** Overexpression of miR-210 in *miR-210Δ; GMR-Gal4>UAS-miR-210* flies was verified by qRT-PCR (**$P < 0.01$; one-way ANOVA, $n = 3$). **(B)** ERG showed that the receptor potential in *miR-210Δ; GMR-Gal4>UAS-miR-210* flies was significantly increased compared with the UAS and Gal4 driver control in the *miR-210Δ* background in 4-d-old flies (*miR210Δ; GMR-Gal4>UAS-miR-210* versus *miR-210Δ; GMR-Gal4/+* or versus *miR-210Δ*; and *UAS-miR210/+*: ***$P < 0.001$, one-way ANOVA, followed by Tukey post hoc test). *miR-210Δ; UAS-miR-210/+* control flies already showed a slightly increased receptor potential (*$P < 0.05$; one-way ANOVA, $n = 7$). **(C)** Toluidine-stained semi-thin sections (upper panel) and (lower panel) TEM demonstrated that the number of rhabdomeres (arrow) increased in both *miR-210Δ; UAS-miR-210/+* control flies and *miR-210Δ; GMR-Gal4>UAS-miR-210* mutants (4 d old). The number and size of vacuoles (arrowheads) decreased in comparison with *miR-210Δ; GMR-Gal4/+* control flies.

degeneration is independent of light and apoptosis and that other mechanisms must underlie the observed phenotype.

### Identification of putative miR-210 targets by RNA sequencing

To address which mechanisms might underlie *miR-210*–dependent degeneration of photoreceptor cells, we performed RNA sequencing analysis, by comparing the expression profiles of *miR-210Δ* null mutants with control flies to identify potential miR-210 target genes. Therefore, we used heads of 1-h-old *miR-210Δ* mutants (n = 3), a time point where photoreceptor cells showed only mild morphological differences but were still functional (Fig 3A–D). As miRNAs are known to down-regulate their targets, we expected direct miR-210 target genes to be up-regulated in *miR-210Δ* mutant flies. The RNA sequencing, resulted in the detection of approximately 8,500 genes of which 812 were differentially regulated between *miR-210Δ* mutant and control flies (Supplemental Data 2). Most differentially regulated genes (509) were down-regulated in *miR-210Δ* mutants, but 303 genes were up-regulated (Fig 5A). Gene ontology (GO) term analysis showed a strong enrichment for genes involved in phototransduction and rhabdomere function, consistent with our hypothesis that *miR-210* is

essential for vision (Fig 5B). Furthermore, up-regulated genes were enriched for the GO term fatty acid biosynthetic process and lipid metabolic process (Fig 5B). Lipid signalling is important for the phototransduction cascade, as G-protein–coupled hydrolysis of the phospholipid phosphatidylinositol 4,5-bisphosphate plays a key role in signal transduction upon light stimulation (Raghu et al, 2012). Notably, we analysed the gene expression changes in the whole head of flies and not specific for photoreceptor cells, suggesting that changes in expression in photoreceptors are underestimated. The differences in the transcriptome of 1-h-old *miR-210Δ* flies indicate the cells had already started to change at the molecular level, although we detected only mild morphological differences at this age based on TEM (Fig 3D). These results might suggest that the retinal degeneration observed in *miR-210Δ* is extremely fast or might already start during late pupae development.

Diacylglycerol kinase (Dgk) showed the strongest up-regulation, with an induction of seven to eightfold in *miR-210Δ* mutant flies (n = 3, ***$P < 0.001$) (Figs 5A and S5A), and TargetScan analysis (Lewis et al, 2005) identified two miR-210-3p and one miR-210-5p* seed matches in the open reading frame of Dgk. Furthermore,

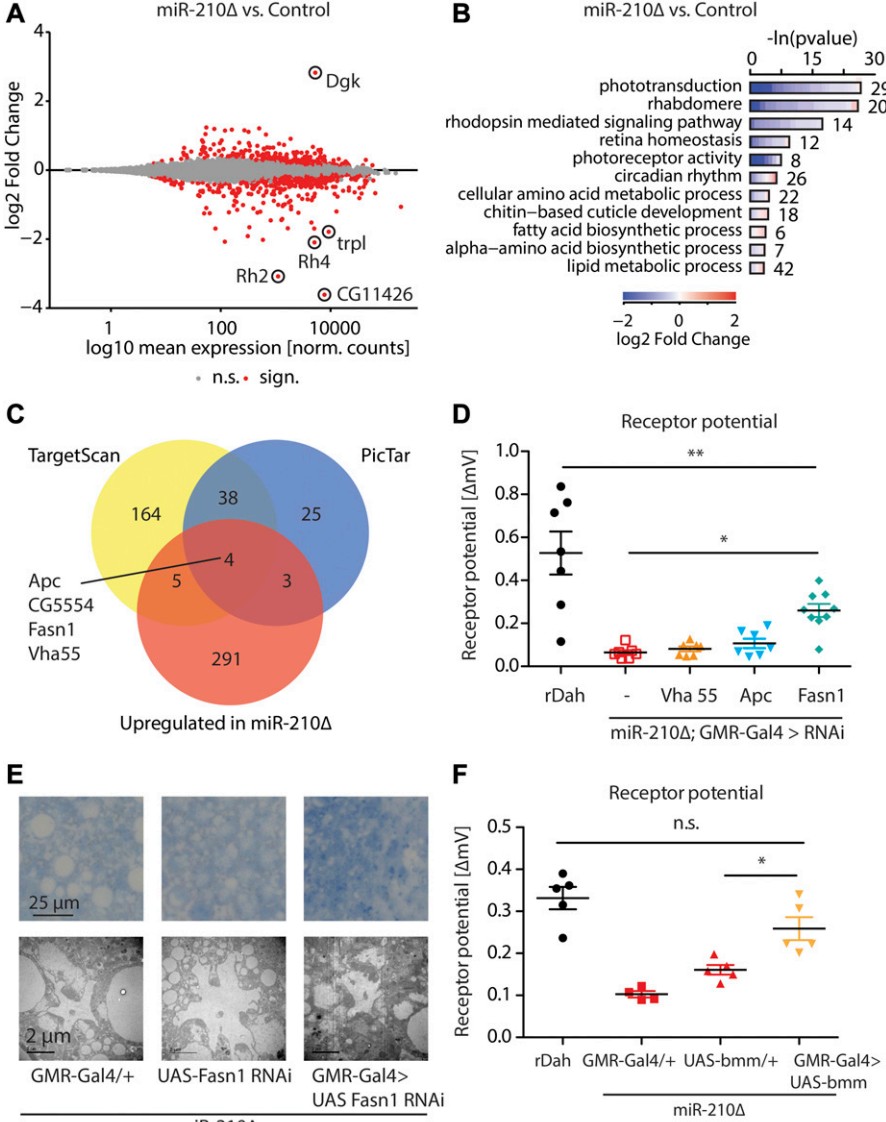

**Figure 5. Transcriptome analysis of *miR-210Δ* mutant flies revealed potential miR-210 targets.**
**(A)** 812 differentially expressed RNAs were identified in 1 h old *miR-210Δ* heads compared with control by RNA sequencing (*adjusted *P* value < 0.05, n = 3). **(B)** GO enrichment of differentially regulated mRNAs demonstrated that many genes with GO terms related to phototransduction and vision were down-regulated. **(C)** Four potential miR-210 targets predicted by PicTar and TargetScan were also up-regulated on RNA level in *miR-210Δ* mutant flies. **(D)** The receptor potential of *miR-210Δ* mutants was partially rescued by Fasn1 RNAi as measured by ERG in 4-d-old flies (*miR-210Δ; GMR-Gal4/+* versus *miR-210Δ*; and *GMR-Gal4>UAS-Fasn1 RNAi*: **P* < 0.05, n = 7–9). **(E)** Photoreceptors in *miR-210Δ; GMR-Gal4> Fasn1 RNAi* showed a similar grade of retinal degeneration compared with *miR-210Δ* controls. **(F)** ERG showed that the receptor potential of *miR-210Δ* mutants was significantly increased by overexpression of bmm lipase (*miR-210Δ; GMR-Gal4/+* versus *miR-210Δ; GMR-Gal4>UAS-bmm*: ***P* < 0.001; *miR-210Δ; UAS-bmm/+* versus *miR-210Δ*; and *GMR-Gal4>UAS-bmm*: **P* < 0.05, n = 4–5).

diacylglycerol kinases such as the *retinal degeneration gene A* (*rdgA*) have been implicated in retinal degeneration in the fly (Masai et al, 1992, 1993, 1997; Harden et al, 1993), making Dgk a prime candidate to mediate retinal degeneration upon loss of *miR-210*. However, knockdown of Dgk in the *miR-210Δ* mutant background did not rescue retinal degeneration, and overexpression of Dgk did not induce retinal degeneration in *miR-210Δ* mutants (Fig S5B–G). These results might suggest that Dgk is not a causal target in retinal degeneration or that miR-210 targets several proteins that have to act in concert to cause retinal degeneration in *miR-210Δ* flies. To identify direct targets of miR-210, we combined the RNA sequencing analysis with computer-based prediction algorithms (Grun et al, 2005; Lewis et al, 2005). As most algorithms produce many false-positive hits, we used a more stringent analysis by overlapping the predicted miR-210 targets by PicTar and TargetScan and generated a set of 42 high-confidence miR-210 targets that were predicted by

both algorithms. Comparison of the 42 miR-210 targets with our RNA sequencing data allowed us to identify four putative miR-210 targets: Apc, CG5554, Fasn1, and Vha55 (Fig 5C). One or more of these genes may, therefore, be direct miR-210 targets in vivo. To test this, we used transgenic RNAi to knockdown Apc, Vha55, and Fasn1 in the *miR-210Δ* mutant background and measured the electro-retinograms (ERG) of these flies at 4 d of age. Interestingly, RNAi-mediated knockdown of fatty acid synthase 1 (Fasn1) but not of Apc or Vha55 partially rescued the ERG defects of *miR-210Δ* mutant flies (n = 5-7; **P* < 0.05) (Fig 5D), suggesting that up-regulation of Fasn1 activity might contribute to the retinal degeneration observed in *miR-210Δ* mutants. Notably, photoreceptor morphology in Fasn1 RNAi rescue flies was still abnormal, verifying that knockdown of Fasn1 RNAi can only partially rescue *miR-210*–dependent retinal degeneration. Next, we overexpressed Fasn1 in the fly eye of wild-type flies using a previously published *UAS-Fasn1* mutant line

(Garrido et al, 2015), to test if overexpression of Fasn1 would be sufficient to cause retinal degeneration. However, ERGs of Fasn1 overexpression flies did not show any abnormalities even in 10-d-old flies (n = 4-5) (Fig S6A), demonstrating that Fasn1 up-regulation alone was not sufficient to cause retinal degeneration, which suggests the contribution of further genes to the observed phenotype. To test whether miR-210 is able to bind and degrade Fasn1 directly, we performed an in vitro luciferase assay targeting the 3′-UTR of Fasn1. However, we were not able to detect an *miR-210*–dependent decrease in luminescence when compared with a scrambled control (Fig S6B). Thus, it is currently not clear whether Fasn1 is a direct target of miR-210 in vivo or whether the regulation of Fasn1 is part of a toxic downstream mechanism. Down-regulation of Fasn1 is expected to result in decreased lipogenesis. To test whether we would also be able to rescue the phenotype of *miR-210* mutants by increasing lipolysis, we overexpressed the triacylglyceride lipase Brummer (Bmm) (Grönke et al, 2005) in *miR-210Δ* mutants. Similar to the knockdown of Fasn1, overexpression of Bmm partially rescued *miR-210Δ*–induced retinal degeneration as measured by ERG (n = 4-5, ***P < 0.001) (Fig 5F), indicating lipid accumulation as a potential downstream mechanism mediating the toxicity upon lack of *miR-210* in the fly retina. Interestingly, lipid metabolism has previously been implicated in retinal degeneration in flies, for example , the fatty acid transporter protein (FATP) is important for lipid homeostasis in the retina and is essential for photoreceptor survival (Dourlen et al, 2012, Van Den Brink et al, 2018). Furthermore, mitochondrial dysfunction leads to glial lipid accumulation and ultimately retinal degeneration (Liu et al, 2015). Lipid metabolic processes were identified as an enriched GO term in our RNA sequencing experiment, which is consistent with the hypothesis that altered lipid homeostasis, might contribute to the retinal degeneration observed in *miR-210Δ* mutant flies.

In summary, we demonstrated that miR-210 is highly specifically expressed in the fly eye and loss-of *miR-210* leads to strong retinal degradation with age, accompanied by changes in the transcriptome already at young age. Our results suggest that miR-210 plays a crucial role in photoreceptor maintenance and that a disrupted lipid homeostasis might cause the *miR-210Δ*–dependent retinal degeneration.

The functional seed sequence "UGCGUGU" of miR-210 is 100% identical between flies, mice, and humans, but interestingly miR-210 is not present in *C. elegans* (Griffiths-Jones et al, 2006), which lacks an eye. The high evolutionary conservation might suggest that the function of miR-210 in photoreceptor maintenance could be conserved from flies to mammals. Genome-wide expression studies indicated previously that miR-210 is enriched in the mouse eye (Xu et al, 2007; Hackler et al, 2010; Karali et al, 2016), similarly to the photoreceptor-specific expression of miR-210 that we observed in the fly. The eye-specific expression of miR-210 in mice raises the possibility that its function in the eye is evolutionarily conserved between flies and mice. In mice, other eye-specific miRNAs such as the miR-183/96/182 cluster have been shown to play a crucial role in photoreceptor maintenance (Lumayag et al, 2013; Busskamp et al, 2014), demonstrating that miRNAs indeed are essential for vision in mammals. Furthermore, previous studies have demonstrated that miR-210 is expressed in the mouse retina and is induced in light-adapted compared with dark-adapted mouse retinas (Krol et al,

2010). miR-210 was also up-regulated in a mouse model of oxygen-induced retinopathy (Liu et al, 2016), linking the expression of miR-210 in the eye to oxygen. Although miR-210 expression in the human eye was not detected in a recent deep sequencing study (Karali et al, 2016), miR-210 was identified as an eye-expressed miRNA in another study (Ragusa et al, 2013). Furthermore, miR-210 has been associated with age-related macular degeneration (AMD), a human disease that leads to blindness with age. In a genome-wide association study in humans, a single-nucleotide polymorphism was identified in the miR-210–binding site of Complement Factor B (Ghanbari et al, 2017). This single-nucleotide polymorphism caused reduced miR-210 binding and increased level of its target complement factor B, which is a known player in AMD, thereby possibly contributing to the AMD disease mechanism. Strikingly, hypoxia and angiogenesis are heavily involved in AMD (Blasiak et al, 2014), linking the function of miR-210 in hypoxia and angiogenesis in mammals to the retinal degeneration we observed in *miR-210Δ* mutant flies. In the future, it will be essential to investigate the role of miR-210 in mammalian eye function. *miR-210 KO* mice are available and have been used in studies investigating the immune system (Mok et al, 2013; Wang et al, 2014) and pulmonary hypertension (White et al, 2015). However, the eye function of *miR-210 KO* mice has not yet been investigated.

In summary, we demonstrated for the first time a crucial role for miR-210 in the function and morphology of the fly eye. As miR-210 is also expressed in the retina of mice, it is tempting to speculate that miR-210 might also fulfil a similar role in the mammalian eye.

## Materials and Methods

### Maintenance of flies

Fly stocks were kept at 25°C on a 12-h light and 12-h dark cycle and fed a standard sugar/yeast/agar diet (Bass et al, 2007). The light intensity in the fly chambers was around 1,000 lux. The flies were reared at controlled larval densities, and once-mated female flies were used for all experiments unless otherwise stated. The flies were snap-frozen with liquid nitrogen. Dissections were carried out in PBS and tissues either directly analysed or frozen on dry ice.

Transgenic flies were backcrossed into the outbred white Dahomey (wDah) or red Dahomey (rDah) wild-type strain (Grönke et al, 2010) with the endosymbiont Wolbachia for at least six generations, if necessary. The following transgenic fly lines were used in this study: *GMR-Gal4* (Bloomington), *UAS-miR-210* (Bejarano et al, 2012), *Sima KO/Tm3Sb* (Bloomington #14640), *Dgk RNAi* (Bloomington #41944), *GMR-p35* (Bloomington #5774), Df(1)BSC352 (Bloomington #24376), *Apc RNAi* (VDRC v51468), *Vha55 RNAi* (VDRC v46553), *Fasn1 RNAi* (VDRC v29349) (Dietzl et al, 2007), *UAS-Fasn1* (Garrido et al, 2015), and *UAS-bmm-eGFP* (Grönke et al, 2005).

### Generation of transgenic fly lines

We used the CRISPR/Cas9 system according to previous publications (Port et al, 2014) to generate *miR-210Δ*, *miR-210Δ GFP*, and

*miR-210ΔSeed* mutants. To generate *miR-210Δ* and *miR-210Δ GFP* mutants, first, transgenic flies expressing two guideRNAs upstream and downstream of the *miR-210* gene were generated (pCFD4-miR-210). To generate *miR-210Δ* null mutants, *pCFD4-miR-210* flies were crossed with flies expressing Cas9 and progeny screened by PCR. To generate *miR-210Δ GFP* reporter flies, *pCFD4-miR-210* flies were crossed with flies expressing Cas9, and their progeny was injected with the pBS-miR-210Δ GFP donor template. PCR screening allowed the identification of GFP knockin. To generate *miR-210ΔSeed*, transgenic flies expressing one guideRNA targeting the functional seed sequence of *miR-210* was generated (*pCFD3-miR-210*). Next, *pCFD3-miR-210* flies were crossed with flies expressing Cas9 and progeny screened by PCR.

*UAS-Dgk* flies were generated by cloning the Dgk cDNA into the pUAST attb vector. pUAST attb Dgk was inserted into the fly genome by the *φ*C31 and attP/attB integration system (Bischof et al, 2007) using the attP40 landing site. Primers used for cloning are shown in Table S1.

## Lifespan analysis

Once-mated flies were transferred to vials (10–25 flies/vial). Three times a week, the flies were transferred to fresh vials and deaths scored. Standard SYA (sugar-yeas-agar) food was used for the whole experiment.

## Electroretinography

Fly photoreceptor function was assessed by ERG. The flies were immobilized on a wax block lying on their back. A reference electrode was inserted into the thorax and a second electrode into the fly retina. The background light was reduced during the whole experiment and the flies' eye was stimulated with white light for 2 s. Recordings were done using the Axoclamp 900A amplifier (Molecular Devices) and Digidata 1440A digitizer (Molecular Devices). Notably, for white-eyed mutant wDah, we used wDah as control and for red-eyed mutant rDah, flies were used as control, as the eye colour is known to affect the shape and size of the ERG (Wu & Wong, 1977).

## RNA extraction and cDNA synthesis

Total RNA including miRNAs was isolated with the miRNeasy Mini Kit (QIAGEN) following the animal tissue protocol. RNA concentration was measured by the Qubit BR RNA assay (Thermo Fisher Scientific). cDNA of mRNA was generated using the SuperScript III first-strand synthesis kit (Invitrogen) using random hexamers. 600 ng of total RNA was used for cDNA synthesis. cDNA synthesis of miRNAs was performed using the TaqMan MicroRNA Reverse Transcription Kit (Life Technologies) using specific primers for each small RNA. 300 ng of RNA was used for cDNA synthesis.

## qRT-PCR

For qRT-PCR of mRNA, PowerUp SYBR Green Master Mix (Thermo Fisher Scientific) or TaqMan Universal Real-Time PCR Master Mix (Life Technologies) was used according to the manufacturer's manual. For qRT-PCR of miRNAs, TaqMan Universal Real-Time PCR Master Mix (NoAmpEraseUNG; Life Technologies) and miRNA-specific TaqMan assays were used. qRT-PCR was performed with the 7900HT real-time PCR system (Applied Biosystems) or with the QuantStudio7 real-time PCR system (Thermo Fisher Scientific). Relative expression (fold induction) was calculated using the ΔΔCT method, and Rpl32 (for mRNAs) or snoRNA442 (for miRNAs) used as a normalization control. Primers used for qRT-PCR are summarized in Table S2.

## RNA sequencing of miRNAs

For miRNA sequencing, we dissected brains, thorax (thorax without the gut), fat body (abdomen without the gut and ovaries), and the gut (midgut without malphigian tubules). Small RNA was enriched from 2 μg total RNA by gel electrophoresis using a Bio-Rad PROTEAN II xi Cell with a 15% polyacrylamide/urea gel in 0.5× TBE buffer. The samples were mixed with loading dye (2×; 89.75% formaldehyde, 10% TBE [5×], 0.05% sodium dodecyl sulfate, and 0.05% bromo-phenol blue) and run at 300 V for 300 min. Small RNA between 19 and 26 bp was cut out and used for library preparation. Small RNA sequencing libraries were generated using the Small RNA v1.5 Kit (Illumina), following the manufacturer's protocol. RNA sequencing was performed with an Illumina HighSeq2500, single-end reads and 100-bp read length at the Max Planck Genome Centre Cologne (Germany). miRNA reads were identified using miRDeep (Friedlander et al, 2008) and miRNAs with a minimum read number of 10 were included for further analysis. The tissue-specificity score was calculated as described previously (Landgraf et al, 2007) and miRNAs with a tissue-specificity score >1 were defined as tissue specific. The data have been deposited in NCBI's Gene Expression Omnibus (Edgar et al, 2002) and are accessible through (GEO: GSE118004).

## RNA sequencing of mRNA

Total RNA was extracted by Trizol (Thermo Fisher Scientific) following standard protocols. Poly (A) capture libraries were generated at the Max Planck Genome Centre Cologne (Germany). RNA sequencing was performed with an Illumina HighSeq2500 and 25 million single-end reads/sample and 150-bp read length at the Max Planck Genome Centre Cologne (Germany). Raw sequence reads were quality-trimmed using Trim Galore! (v0.3.7) and aligned using Tophat2 (Kim et al, 2013) (v2.0.14) against the Dm6 reference genome. Multi-mapped reads were filtered using SAMtools (Li et al, 2009). Data visualization and analysis was performed using SeqMonk, custom RStudio scripts, and the following Bioconductor packages: Deseq2 (Love et al, 2014), topGO, ReactomePA, and org. Dm.eg.db. For visualization of functional enrichment analysis results, we further used the CellPlot package. The data have been deposited in NCBI's Gene Expression Omnibus (Edgar et al, 2002) and are accessible through (GEO: GSE118004).

## Immunostainings of *Drosophila* tissues

Heads without the proboscis or manually dissected tissues were dissected in PBS and fixed with 4% paraformaldehyde at 4°C for 2 h. For cryosections, heads were incubated in 30% sucrose at 4°C overnight, mounted in TissueTek, and cut into 10-μm thin sections.

Cryosections or manually dissected tissues were washed 6 × 30 min in PBT (PBS with 0.5% Triton X-100) and subsequently blocked in 1 ml blocking buffer (PBT with 5% fetal bovine serum and 0.01% sodium azide) for 60 min. The following dilutions of primary antibodies were used for incubation over night at 4°C: 1:200 anti-chaoptin (24B10; DSHB); 1:1,000 anti-GFP (A10262; Life Technologies). Following washes in PBT, the tissues were incubated with a suitable Alexa Fluor secondary antibody (Molecular Probes) overnight at 4°C. Following washes in PBT, the tissues were incubated in 50% glycerol in PBS for 30 min and subsequently mounted on a microscope slide in VectaShield Antifade Mounting Medium with DAPI (Vectorlabs). Imaging was done using a Leica SP5-X or Leica SP8-X confocal microscope.

### TEM

TEM was done according to (Johnson et al, 2002) with modifications. In brief, fresh fly heads were cut in half and first fixed in 25% glutaraldehyde in $dH_2O$ (Sigma-Aldrich) for 1 h at RT, then in 1% osmium/2% glutaraldehyde in the dark on ice for 30 min, followed by 1 h fixation in 2% glutaraldehyde in the dark on ice. After washing and dehydration in EtOH, the heads were incubated 2 × 10 min in 100% EtOH, 2 × 10 min in acetone, and in 1:1 acetone:epon (Sigma-Aldrich) overnight at 4°C. Eyes were mounted in epon (Sigma-Aldrich) and polymerized at 65°C for 72 h. Ultrathin sections (70 nm) were contrasted in 1.5% uranylacetate for 15 min, followed by lead nitrate solution (1.3 M sodium citrate, 1 M lead nitrate, and 1 M sodium hydroxide) in a $CO_2$-free environment. Images were acquired using JEM 2100Plus TEM (JEOL).

During sample preparation for TEM, after dehydration, several eyes were embedded in araldite, and 2.5-$\mu$m semi-thin sections were taken with a Reichert OM U2 microtome, followed by toluidine staining to obtain a better overview of the whole retina.

### Luciferase assays

Human HEK293T cells were used for the luciferase-based miRNA target validation experiments. Transfection of pMIR REPORT including Fasn1 3′-UTR attached to the firefly luciferase ORF, pRL (Renilla luciferase), and miRVana miRNA mimics (dme-miR-210-3p or Negative Control #1; Thermo Fisher Scientific) was achieved by using Lipofectamine 3000 (Thermo Fisher Scientific) according to the manufacturer's manual. The Dual-Glo Luciferase Assay System (Promega) was used to quantify firefly and Renilla luciferase.

### Statistical analysis

Statistical analysis was performed using GraphPad Prism. Individual statistical tests are mentioned in the respective figure legends. One-way ANOVA was always followed by Tukey post hoc test. Two-way ANOVA was always followed by Bonferroni post hoc test. Lifespan assays were recorded using Excel, and survival was analysed using log rank test. Significance was determined according to the $P$-value: $*P < 0.05$, $**P < 0.01$, $***P < 0.001$, and $****P < 0.0001$.

## Supplementary Information

## Acknowledgements

We thank Christian Kukat and the FACS and imaging core facility at the Max Planck Institute for Biology of Ageing for help with the microscopy data, as well as Astrid Schauss and Janine Klask from the Imaging Core Facility at the CECAD for their support in generating the electron microscopy data. Furthermore, we acknowledge the Bioinformatics core facility at the Max Planck Institute for Biology of Ageing for their help with the analysis of high throughput data and the Max Planck Genome Centre Cologne for generation of sequencing libraries and performing next generation sequencing. We would like to thank Isabelle Schiffer for providing HEK293T cells, pMIR Report, and pRL plasmids. We would also like to acknowledge Ferdinand Grawe for his support with the electron microscopy and Michael Dübbert for his support with the electrophysiology. We would like to thank Oliver Hendrich for technical and organisational assistance and the whole Partridge lab for helpful feedback. We are grateful to Prof Eric C Lai for supplying the *UAS-miR-210* fly line and Jacques Montagne for the *UAS-Fasn1* fly line. Stocks obtained from the Bloomington *Drosophila* Stock Center (NIH P40OD018537) and the Vienna Drosophila Resource Center were used in this study. The 24B10 (chaoptin) antibody developed by the California Institute of Technology was obtained from the Developmental Studies Hybridoma Bank, created by the National Institute of Child Health and Human Development of the National Institutes of Health and maintained at The University of Iowa, Department of Biology, Iowa City, IA 52242.

### Author Contributions

CM Weigelt: conceptualization, data curation, formal analysis, investigation, and writing—original draft, review, and editing.
O Hahn: data curation and visualization.
K Arlt: data curation.
M Gruhn: methodology.
AJ Jahn: data curation.
J Eßer: methodology.
JA Werner: data curation.
C Klein: methodology.
A Büschges: methodology.
S Grönke: conceptualization, supervision, investigation, project administration, and writing—original draft, review, and editing.
L Partridge: conceptualization, supervision, funding acquisition, project administration, and writing—original draft, review, and editing.

### Conflict of Interest Statement

The authors declare that they have no conflict of interest.

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
