## [Reviewer comments · Life Science Alliance]

Life Science Alliance

Loss of miR-210 leads to progressive retinal degeneration in *Drosophila melanogaster*

Carina Weigelt, Oliver Hahn, Katharina Arlt, Matthias Gruhn, Annika Jahn, Jacqueline Eßer, Jennifer Werner, Corinna Klein, Ansgar Büschges, Sebastian Grönke, and Dame Partridge

DOI: [10.26508/lsa.201800149](https://doi.org/10.26508/lsa.201800149)

Corresponding author(s): Sebastian Grönke, Max-Planck Institute for Biology of Ageing and Dame Partridge, Max-Planck Institute for Biology of Ageing

Review Timeline:

Submission Date:	2018-08-08
Editorial Decision:	2018-08-31
Revision Received:	2018-11-30
Editorial Decision:	2019-01-07
Revision Received:	2019-01-10
Accepted:	2019-01-11

Scientific Editor: Andrea Leibfried

Transaction Report:

August 31, 2018

Re: Life Science Alliance manuscript #LSA-2018-00149-T

Dr. Sebastian Grönke
Max-Planck Institute for Biology of Ageing
Biological Mechanisms of Ageing
Joseph-Stelzmann Str. 9b
Cologne, NRW 50931
Germany

Dear Dr. Grönke,

Thank you for submitting your manuscript entitled "Loss of miR-210 leads to progressive retinal degeneration in *Drosophila melanogaster*" to Life Science Alliance. The manuscript was assessed by expert reviewers, whose comments are appended to this letter.

As you will see from the reports, the referees appreciate that your data present a valuable resource to the field and that your analysis is the first to link miR-210 to retinal maintenance. However, you will also see that the referees find that several aspects of the study would need to be strengthened before they can support publication here. From our side, we realise that most miRNAs have numerous targets and that providing a functional rescue via a single target is not feasible in many cases. Identifying the responsible miR210 targets will therefore not be a requirement for publication.

Given the overall interest from the referees we would like to invite you to submit a revised version of your manuscript to Life Science Alliance.

For more detail on the exact requirements for the revision, please see the annotated version of the referee reports (attached as a separate word document here).

Thank you for this interesting contribution to Life Science Alliance. We are looking forward to receiving your revised manuscript.

Sincerely,

Anne Nielsen PhD
Senior Editor

Andrea Leibfried
Executive Editor

Life Science Alliance

- A letter addressing the reviewers' comments point by point.
- An editable version of the final text (.DOC or .DOCX) is needed for copyediting (no PDFs).
- High-resolution figure, supplementary figure and video files uploaded as individual files: See our detailed guidelines for preparing your production-ready images, <http://life-science-alliance.org/authorguide>
- Summary blurb (enter in submission system): A short text summarizing in a single sentence the study (max. 200 characters including spaces). This text is used in conjunction with the titles of papers, hence should be informative and complementary to the title and running title. It should describe the context and significance of the findings for a general readership; it should be written in the present tense and refer to the work in the third person. Author names should not be mentioned.

B. MANUSCRIPT ORGANIZATION AND FORMATTING:

Full guidelines are available on our Instructions for Authors page, <http://life-science-alliance.org/authorguide>

Reviewer #1 (Comments to the Authors (Required)):

The paper by Carina M. Weigelt et al. entitled "Loss of miR-210 leads to progressive retinal degeneration in *Drosophila melanogaster*" submitted to the Life Science Alliance Journal contains the primary data describing the results of a small RNA sequencing screen that aimed to identify tissue-specific miRNAs in *Drosophila*. Recently, multiple similar screens profiling developmental-stage or tissue-specific miRNAs have been performed and published. This work adds new information about the distribution of miRNAs in metabolically important tissues, such as the adult brain, thorax, gut, and fat body. Then, one of the most brain-specific miRNAs, miR-210, was analyzed in greater detail. To do so, the authors generated a miR-210 loss of function mutant and a reporter line.

Unfortunately, the authors planned their experiments trying to confirm the conserved role of this miRNA in hypoxia based on previously published findings in vertebrates. However, it appeared that the function of miR-210 in the response to hypoxia might not be conserved between flies and mammals. Instead, they found that miR-210 is expressed in photoreceptors, ocelli, and the antennal lobes. Loss of miR-210 caused age-dependent loss of photoreceptor integrity and reduced photoreceptor function, which was measured by electroretinography. All the experiments performed are convincing and well done, and the conclusions are clear; however, the manuscript leaves the impression of being unfinished and even somewhat sad, since all the conclusions are based mostly on negative results. It is a common practice for miRNA research studies to go all the way from the miRNA phenotype to finding the relevant target, increased expression of which would be causative for the described phenotype. Here, the authors could not prove that *Dgk* is relevant for the photoreceptor degeneration phenotype. Then, by doing elaborate computational analysis, they found 4+8 additional putative miR-210 targets. Strangely, they have not tested any of them, which is my major concern about this work.

To make this story complete, it would be important:

- to test via qPCR the expression levels of at least the four targets which were predicted by both databases (*Apc*, CG5554, *Fasn 1*, *Vha55*),
- to see whether overexpression of any of them mimics the miR-210 lof phenotype and if so,
- try to rescue miR-210 photoreceptor phenotype by downregulation of the target using RNAi or heterozygous mutants.

Minor concerns

1. There is a little discrepancy when showing the representative images of photoreceptors in miR-210 mutants. Even though it was mentioned that the EM detects degeneration earlier than confocal, in Figure 3A (day 10), photoreceptor cells are clearly present in miR-210 mutants (they are abnormally arranged but most of them are still there), while in Figure 3D (day 10), ALL photoreceptor cells are degenerated. If this phenotype is so variable, it should either be quantified or authors should show images that better represent the phenotype. I think that quantification would be preferable, but not obligatory; it would not change the story.

2. Since miR-210 is highly expressed in the brain of ADULT flies and most of the experiments were done on adults, it would make sense to test the survival of adult miR-210 animals under the hypoxic condition. Currently, only the percentage of flies eclosing under hypoxia has been quantified. Thus, the possible role of miR-210 under hypoxia was addressed only during development, but not during adulthood and ageing.

3. In the sentence "In summary, we demonstrated for the first time that miR-210 is highly specifically expressed in the fly sensory organs...", the phrase "for the first time" should be deleted, since the miR-210 expression pattern has been already published.

In summary, the data described in this manuscript are produced in a methodologically sound

manner. This manuscript would be useful and important for researchers with closely related interests, and might provide a good list of tissue-specific miRNAs for further investigation. If the raised concerns were addressed, I would support publication of the revised manuscript in the Life Science Alliance.

Reviewer #2 (Comments to the Authors (Required)):

1. A short summary of the paper

The authors set out to identify small RNAs which are expressed in a tissue-specific way. Therefore, they did RNA seq from adult brain, thorax, gut and fat body. They concentrated further analysis on miR-210, which had recently been shown to be responsive to hypoxia in mammalian cells. They created a CRISPR-induced deletion of the miR-210 locus. Homozygous mutant flies are viable and fertile, but not sensitive to hypoxia. Mutant flies show progressive loss of photoreceptor integrity, which was associated with loss of photoreceptor function as measured by ERG. Finally, some results are shown to suggest downstream targets of miR-210.

Topic-wise, this manuscript could potentially be interesting for people working on miRNAs and/or on retinal degeneration. However, my impression is that the submission was premature, since some of the data shown are not convincing, and several of the conclusions drawn from the results are not justified by the data.

2. For each main point of the paper, please indicate if the data are strongly supportive.

a. Identification of tissue-specific miRNAs

- The origin of the RNAs used for sequencing is not at all clear. What does "brain" mean? Are these dissected brains or just heads? Similarly, what does "gut" mean, midgut, hindgut?
- I am somewhat puzzled by the statement that miR-210 is highly conserved. According to Flybase, no orthologues are reported.
- Fig. 1C: the stages are purely specified. Were the RNAs from whole animals? Or from brains?

b. miR-210 is not essential for survival under hypoxic conditions

- Fig. EV1A, B: RNA from heads? Brain? Whole flies? What is wDah?
- Fig. EV1C: I guess it should be "% larvae hatching"?
- Fig. EV1E: A recent paper (Chen et al., not cited) showed that the lifespan of miR-201 deficient males is reduced.

c. Loss of miR-210 leads to retinal degeneration

- The authors showed that miR-210 Delta-GFP is expressed in the "optic lobes, the ocelli and the antennal lobes, which are all important sensory organs". The optic lobes are not sensory organs. In addition, I am not convinced about the identity of the ocelli. Furthermore, the authors do NOT show any expression of miR-210 in photoreceptor cells! And in the paper by Cusumano, which they cite, it is also not convincing that there is expression in photoreceptor cells (could also be pigment cells).
- Fig. 3: I am not convinced that "miR-210 is not essential for photoreceptor development", as stated by the authors. The TEM pictures clearly reveal defects in rhabdomere organization! And even at 0 hour (Fig. 3D), it looks that one of the cells is already undergoing apoptosis (darkly stained cytoplasm). It is also not clear whether photoreceptor disappear or just the rhabdomeres, which would go along with loss of Chaoptin staining. In Fig. 3B, many nuclei are still visible, even at 42 days.
- Fig. EV2: From this figure I would assume that there is a defect upon overexpression of miR-210.

To exclude any defect, TEM pictures should be provided.

- Fig. 4B I do not understand why they plot "Receptor potential [Delta mV]" here and "Receptor potential [mV]" in Fig. 3c.
- Fig. 4C: I think that overexpression of miR-210 results in a split rhabdomere, rather than an additional rhabdomere.
- The authors demonstrated the retinal phenotype of miR-210 in homo- or hemizygous flies (it should be specified, which flies they analysed). Since the rescue experiment showed only partial rescue, they should provide additional data that the defects are due to loss of miR-210, for example by looking at miR-210 Delta/deficiency.
- The defects observed in miR-210 mutants could also be due to lack of miR-210 in the optic lobes, rather than in photoreceptors (in particular, since expression in photoreceptors was not shown). Therefore, the authors could induce clones in the retina and study their phenotype.

d. miR-210-mediated retinal degeneration is independent of light and apoptosis

- They write "As miR-210 is expressed in the fly eye ...". This has not been shown.
- The light conditions (Lux) should be specified in Materials and Methods.

e. Identification of putative miR-210 targets

- Fig. EV4E: for me it appears that knocking down Dgk could partially rescue the miR-210 phenotype since more rhabdomeres are visible, at least in this section. In general, I recommend to also show an overview of the retina, to get an idea about the variability of the phenotype.
- Fig. 5 is missing.

Reviewer #3 (Comments to the Authors (Required)):

Weigelt et al. study miRNA expression in four tissue types of *Drosophila melanogaster*. The authors categorize these hundreds of miRNA molecules according to their specific expression patterns and focus on the brain specific miR-210. This miRNA molecule has previously been associated with hypoxia. However, the authors clearly demonstrate that miR-210 regulation doesn't impact hypoxia in *Drosophila*. In contrast, miR-210 deletion results in degeneration of adult photoreceptors linking regulatory functions to the maintenance of these sensory neurons. This novel miR-210-mediated phenotype is well and plausible described and supported by the data. In addition, the authors aim at providing mechanistic insights, however, that aspect appears rather premature. Some claims are also not well supported by data or cited literature.

Major comments:

- Results about miR-958 and miR-314: The authors only present tissue-specific expression patterns. How is this a functional proof for "gut-specific responses to bacterial infection" or "that miR-314 indeed plays an important function in the gut"? These statements are not supported by the data shown.
- How "highly expressed" is miR-210? Is this based on small RNA-Seq, which is known to have high amplification biases? How high is miR-210 expression in relation to other brain-enriched miRNAs such as miR-124?
- The miR-210 overexpression data during development is interesting and reproduces partly previous work (Cusumano et al. 2018). This could be mentioned in the text. Do the authors also see

more than seven rhabdomeres in wildtype eyes when overexpressed?

- Fig.5 A, B just reflect the phenotype at the transcriptional level, i.e. photoreceptor loss, but do not directly link miR-210 to it. The fact that the highest upregulated gene, Dgk, is a potential miR-210 target but cannot functionally be linked to the phenotype requires a deeper discussion. Either the data is wrong, or Targetscan not accurate, or both.
- In Figure 5C, the authors use two target prediction tools and identify four transcripts with annotated binding sites and upregulation in miR-210 deleted samples. Why do the authors not test and validate these genes (upregulation in wt or knock-down in the miR-210 mutant)?
- Does evolutionary conservation of a miRNA sequence correlate with their function in different species and tissues? This implication is not well justified. The miR-210 phenotype in flies shows characteristics of the miR-182 and miR-183 phenotype in mouse photoreceptors (Buskamp et al. 2014), suggesting that specific functions are mediated by different miRNAs in different species.
- The claim that miR-210 is "one of the highest expressed miRNAs in the mouse retina" is also not well justified. One cannot directly extract expression levels from small RNA-Seq experiments. Krol et al. only mention miR-210 in a supplementary table.
- Linking miR-210 function to human phenotypes has also to be taken with extreme care as miR-210 was not a hit in a human retina study (Karali et al. 2016). Therefore, projecting conserved photoreceptor maintenance functions of miR-210 from flies to mammals is pure speculation as hypoxia via miR-210 was not conserved. The implied "conserved" impact of miR-210 is not plausible.

Minor comments:

- Please show sample numbers 'n' in figures or legend and text and also provide p-values for statistical significance throughout the text.
- Fig.1B: Why is the miR-210 expression only significantly different between brain and fat body?
- Fig.3A, D: For underlying a significant difference between wt and miR-210 knockout, a quantification of the phenomena shown in the images is needed.

Dear Editor,

Thank you for sending us the reviewers' comments on our submission "Loss of miR-210 leads to progressive retinal degeneration in *Drosophila melanogaster*". We were happy to see that all three reviewers found our study interesting.

In the response below, we have addressed all of the remarks raised by the reviewers. Following the advice of reviewers #2 and #3 we added immunostainings pictures of whole mount retinas and cryosections of fly eyes and thereby provide evidence that miR-210 is indeed expressed in the fly retina. Furthermore, as requested by reviewers #1 and #3, we have further functionally characterized the potential target genes, which were upregulated in miR-210 mutants in our RNA seq data set and predicted to be direct targets of miR-210. Interestingly, we were able to demonstrate that the downregulation of one of them, Fatty acid synthase 1 (Fasn1), in the miR-210 mutant background partially rescued the retina degeneration phenotype. Furthermore, we were also able to partially rescue the mutant phenotype by overexpression of the triacylglyceride lipase brummer, demonstrating that both decreased lipogenesis and increased lipolysis are sufficient to rescue miR-210 dependent retina degeneration. Thus, we identified aberrant lipid metabolism as an important factor in toxicity caused by lack of miR-210 in the fly eye. Lipid metabolism has previously been implicated in retinal degeneration in the fly retina, and our data suggest that miR-210 plays an important role in this context. We were not able to pinpoint the regulation of lipid metabolism by miR-210 to a single target gene, however, our RNA seq analysis identified lipid metabolic processes as an enriched term with more than 42 genes regulated, which might suggest that miR-210 has several target genes. We feel that the new data significantly improve the quality of our manuscript and hope that it is now suitable for publication in Life Science Alliance. Please find below a point-by-point response to the reviewer comments.

Reviewer #1:

To make this story complete, it would be important:

- to test via qPCR the expression levels of at least the four targets which were predicted by both databases (Apc, CG5554, Fasn 1, Vha55),

We directly tested the function of *Apc*, *Fasn1* and *Vha55* in miR-210 mediated retinal degeneration *in vivo* to identify potential miR-210 targets. Interestingly, we found that knock-down of *Fasn1* (but not *Apc* or *Vha55*) in miR-210 Δ partially rescued the miR-210 Δ retinal degeneration phenotype (see also below).

- to see whether overexpression of any of them mimics the miR-210 lof phenotype and if so,

As mentioned above, we found that only the knock-down of *Fasn1* partially rescued the miR-210 Δ mediated retinal degeneration phenotype. Therefore, we used UAS-*Fasn1* transgenic flies generated by the Montagne lab to over-express *Fasn1* in the fly eye, to test if the over-expression of *Fasn1* mimics the miR-210 Δ phenotype. However, we did not see any difference in photoreceptor function as measured by ERG in 10 days old flies overexpressing *Fasn1*. These results suggest, that *Fasn1* is required for miR-210 Δ retinal degeneration, but not sufficient to induce the retinal degeneration itself. We believe that additional miR-210 targets might be involved in the retinal degeneration phenotype. We added the *Fasn1* over-expression data to Figure S6A.

- try to rescue miR-210 photoreceptor phenotype by downregulation of the target using RNAi or heterozygous mutants.

As suggested we tried to rescue the miR-210 Δ photoreceptor phenotype by down-regulation of *Vha55*, *Apc* and *Fasn1* using RNAi. Interestingly, we observed a partial rescue of the miR-210 Δ photoreceptor phenotype by down-regulation of *Fasn1*. We added this data and follow-up experiments to the manuscript (compare also the response to reviewer 3). We concluded, that *Fasn1* is partially responsible for the miR-210 Δ -dependent retinal degradation, but that also other targets might contribute to this phenotype. Interestingly, altered lipid metabolism was already linked to retinal degeneration by several other studies. We added the following paragraph to the manuscript: "Furthermore, mitochondrial dysfunction leads to glial lipid accumulation and ultimately retinal degeneration (Liu *et al.* 2015). Lipid metabolic processes were identified as an enriched GO term in our RNA seq experiment, which is consistent

with the hypothesis that altered lipid homeostasis might contribute to the retinal degeneration observed in miR-210 Δ mutant flies."

Minor concerns

1. There is a little discrepancy when showing the representative images of photoreceptors in miR-210 mutants. Even though it was mentioned that the EM detects degeneration earlier than confocal, in Figure 3A (day 10), photoreceptor cells are clearly present in miR-210 mutants (they are abnormally arranged but most of them are still there), while in Figure 3D (day 10), ALL photoreceptor cells are degenerated. If this phenotype is so variable, it should either be quantified or authors should show images that better represent the phenotype. I think that quantification would be preferable, but not obligatory; it would not change the story.

Thank you for highlighting this discrepancy. The EM picture of 10 days old miR-210 Δ mutants still shows some residues of one or more rhabdomer (probably R7). We added semi-thin sections (250 μ m) of toluidine stained eyes at lower magnification to provide a better overview over the whole retina / several ommatidia.

2. Since miR-210 is highly expressed in the brain of ADULT flies and most of the experiments were done on adults, it would make sense to test the survival of adult miR-210 animals under the hypoxic condition. Currently, only the percentage of flies eclosing under hypoxia has been quantified. Thus, the possible role of miR-210 under hypoxia was addressed only during development, but not during adulthood and ageing.

We agree with the reviewer and we now quantified the survival of adult flies on 1 % hypoxia for 14 hours. While none of the heterozygous *sima* mutant flies survived this procedure, miR-210 mutants showed the same survival as wild type flies, demonstrating that miR-210 function is not essential for survival neither during development nor for adult flies. We added these new data to Fig S1D.

3. In the sentence "In summary, we demonstrated for the first time that miR-210 is highly specifically expressed in the fly sensory organs...", the phrase "for the first time" should be deleted, since the miR-210 expression pattern has been already published.

We apologize for this mistake. Only a very recent publication showed the expression pattern and we missed to delete this sentence. We deleted now the phrase “for the first time” and have cited the corresponding manuscript.

Reviewer #2 (Comments to the Authors (Required)):

1. A short summary of the paper

The authors set out to identify small RNAs which are expressed in a tissue-specific way. Therefore, they did RNA seq from adult brain, thorax, gut and fat body. They concentrated further analysis on miR-210, which had recently been shown to be responsive to hypoxia in mammalian cells. They created a CRISPR-induced deletion of the miR-210 locus. Homozygous mutant flies are viable and fertile, but not sensitive to hypoxia. Mutant flies show progressive loss of photoreceptor integrity, which was associated with loss of photoreceptor function as measured by ERG. Finally, some results are shown to suggest downstream targets of miR-210.

Topic-wise, this manuscript could potentially be interesting for people working on miRNAs and/or on retinal degeneration. However, my impression is that the submission was premature, since some of the data shown are not convincing, and several of the conclusions drawn from the results are not justified by the data.

2. For each main point of the paper, please indicate if the data are strongly supportive.

a. Identification of tissue-specific miRNAs

- The origin of the RNAs used for sequencing is not at all clear. What does "brain" mean? Are these dissected brains or just heads? Similarly, what does "gut" mean, midgut, hindgut?

Thank you for pointing out this missing information. We added further details for the dissection of tissues to the material & methods section: “For miRNA sequencing, we dissected brains, thorax (thorax without the gut), fat body (abdomen without the gut and ovaries) and the gut (midgut without malpighian tubules and without the crop).”

- I am somewhat puzzled by the statement that miR-210 is highly conserved. According to Flybase, no orthologues are reported.

The reviewer is right that the evolutionary conservation of miR-210 is not indicated in Flybase. However, for miRNAs, miRbase.org is the commonly used database for sequence and conservation information. miRbase.org clearly shows that miR-210 is conserved between flies and humans and has the identical, functional seed region “UGCGUGU” in all species.

- Fig. 1C: the stages are purely specified. Were the RNAs from whole animals? Or from brains?

We added further information about the stages and tissue in the figure legend: “Up to 20 h old embryos and wandering L3 larvae were used. We used whole animals for this experiment.”

b. miR-210 is not essential for survival under hypoxic conditions

- Fig. EV1A, B: RNA from heads? Brain? Whole flies? What is wDah?

The q-RT-PCR presented in Figure S1A, B was done on RNA isolated from fly heads. We have included this information now in the corresponding figure legend.

wDah is the abbreviation for the outbreed wildtype strain “white Dahomey” that were used in most experiments as control. The wDah abbreviation was mentioned in the material and methods part “Transgenic flies were backcrossed into a white Dahomey (wDah) or red Dahomey (rDah) wild- type strain”. In order to make it more visible we have now introduced the wDah wild type fly line also in the text and cite a relevant reference in the material and method section.

- Fig. EV1C: I guess it should be “% larvae hatching”?

In Fig. S1C we quantified the % of adult flies eclosing. We thank the reviewer for pointing out this missing information and have included this now in the Y-axis label of Figure S1C.

- Fig. EV1E: A recent paper (Chen et al., not cited) showed that the lifespan of miR-201 deficient males is reduced.

In contrast to Chen et al., study, the lifespan of miR-210 mutant males was not reduced in our hands, but rather showed a very small life span extension (** $p < 0.01$). Although we cannot explain the discrepancy between the results of the two studies, the fact that flies were not backcrossed into a common wild type genetic background in the Chen et al., study, which is essential for lifespan studies (Partridge & Gems, 2007), might explain the difference. Another explanation could be the difference in genetic backgrounds used. We have now added the male data (lifespan and starvation resistance) to the manuscript (S1E, F), cited the Chen et al., manuscript and discussed the discrepancy between the two studies.

c. Loss of miR-210 leads to retinal degeneration

- The authors showed that miR-210 Delta-GFP is expressed in the "optic lobes, the ocelli and the antennal lobes, which are all important sensory organs". The optic lobes are not sensory organs. In addition, I am not convinced about the identity of the ocelli. Furthermore, the authors do NOT show any expression of miR-210 in photoreceptor cells! And in the paper by Cusumano, which they cite, it is also not convincing that there is expression in photoreceptor cells (could also be pigment cells).

To address the first criticism we have changed the text to: "miR-210 Δ GFP expression was highly specific to the fly compound eye, the ocelli and the antennal lobes, which are important for sensing of light and olfactory cues."

To further investigate the expression pattern of miR-210 in photoreceptor cells we added immunostaining pictures of whole mount retina and cryosections of whole heads of our miR-210 Δ GFP reporter line. We agree that miR-210 might be expressed not only in the photoreceptor cells, but potentially also in the pigment cells. We added the following sentence to the manuscript: "By whole mount retina stainings as well as cryosection of miR-210 Δ GFP heads, we further demonstrated that miR-210 is also expressed in the fly retina including photoreceptors and potentially pigment cells".

- Fig. 3: I am not convinced that "miR-210 is not essential for photoreceptor development", as stated by the authors. The TEM pictures clearly reveal defects in rhabdomere organization! And even at 0 hour (Fig. 3D), it looks that one of the cells is already undergoing apoptosis (darkly stained cytoplasm). It is also not clear whether photoreceptor disappear or just the rhabdomeres, which would go along with loss of Chaoptin staining. In Fig. 3B, many nuclei are still visible, even at 42 days.

The reviewer is right, that very mild defects are already seen in photoreceptors of very young flies in the TEM analysis, which might indicate problems in photoreceptor development upon lack of miR-210. We have now included a sentence to highlight this possibility "Thus, lack of miR-210 might also mildly affect photoreceptor development."

We believe that the whole photoreceptor cell and not just the rhabdomers are impaired in miR-210 Δ mutant flies, as the cell structure is affected by the appearance of big vesicles, which might be related to autophagy as seen by TEM. It remains to be tested, if the nuclei present at day 42 are from photoreceptor cells or neighbouring cells (e.g. pigment cells).

- Fig. EV2: From this figure I would assume that there is a defect upon overexpression of miR-210. To exclude any defect, TEM pictures should be provided.

Thank you for your suggestion, we added TEM pictures of miR-210 over-expression flies to the new Figure S3. We do not see major changes in photoreceptor arrangement in miR-210 over-expression mutants. However, similarly to miR-210 over-expression in the miR-210 mutant background, we detected a limited number of ommatidia with 8 rhabdomers, which might be split rhabdomers.

- Fig. 4B I do not understand why they plot "Receptor potential [Δ mV]" here and "Receptor potential [mV]" in Fig. 3c.

We apologize for the typing error and changed every figure legend to "Receptor potential Δ mV".

- Fig. 4C: I think that overexpression of miR-210 results in a split rhabdomere, rather than an additional rhabdomere.

We thank the reviewer for the suggestion and have accordingly changed the text to: "However, we noted by TEM that over-expression of miR-210 in the eye led to several ommatidia that presented 8 visible rhabdomers, which might be split rhabdomers (Fig S3 C)."

- The authors demonstrated the retinal phenotype of miR-210 in homo- or hemizygous flies (it should be specified, which flies they analysed). Since the rescue experiment showed only partial rescue, they should provide additional data that the defects are due to loss of miR-210, for example by looking at miR-210 Delta/deficiency.

We used female flies consistently for all experiments concerning the eye phenotype, which is stated in the material and methods section "Flies were reared at controlled larval densities and once-mated female flies were used for all experiments unless otherwise stated." Therefore, miR-210 Δ mutants are homozygous for the miR-210 deletion.

To further confirm that the observed phenotype is indeed caused by lack of miR-210 function, we followed the reviewer's advice and crossed our miR-210 Δ to flies (Df(1)BSC352) that carry a deletion encompassing the miR-210 gene locus. miR-210 Δ / Df(1)BSC352 mutant flies showed the same retina degeneration phenotype as homozygous miR-210 Δ mutant flies. Furthermore, we characterized another independent miR-210 mutant line that we previously generated, which carries a short deletion within the functionally active seed sequence of the miR-210 gene, termed miR-210 Δ Seed. This mutant also shows the retinal degeneration phenotype, demonstrating that the lack of miR-210 is causative for the eye phenotype. We added these data as the new supplemental Figure S2.

- The defects observed in miR-210 mutants could also be due to lack of miR-210 in the optic lobes, rather than in photoreceptors (in particular, since expression in photoreceptors was not shown). Therefore, the authors could induce clones in the retina and study their phenotype.

As mentioned above, we provide now new evidence that miR-210 is indeed expressed directly in photoreceptor cells (Figure 2C, D), which is consistent with the hypothesis that lack of miR-210 in photoreceptor cells causes their degradation. However, we agree with the reviewer that more direct experimental evidence would be required to make this statement. In order to generate clones it is necessary to recombine the mutant allele on an FRT chromosome, which in case of miR-210 is difficult as both the miR-210 gene and the corresponding FRT19A site are in very close proximity to each other on the X chromosome. Unfortunately, we were therefore not able to generate these flies by recombination. In order to address the reviewers concern we added the following sentence to text: “However, the fast degradation of photoreceptor neurons within a few days suggests that miR-210 expression in the photoreceptors, lamina and/or medulla is crucial for the maintenance and function of adult photoreceptor neurons. “

d. miR-210-mediated retinal degeneration is independent of light and apoptosis

- They write "As miR-210 is expressed in the fly eye ... ". This has not been shown.

We provide now new evidence that miR-210 is indeed expressed in the fly eye (Fig. 2C, D).

- The light conditions (Lux) should be specified in Materials and Methods.

The light intensity in the fly chambers was around 1000 lux. We added this information to the Material and Methods part.

e. Identification of putative miR-210 targets

- Fig. EV4E: for me it appears that knocking down Dgk could partially rescue the miR-210 phenotype since more rhabdomeres are visible, at least in this section. IN general, I recommend to also show an overview of the retina, to get an idea about the variability of the phenotype.

This is a very nice suggestion. To each TEM picture, we added Toluidine stainings of semi-thin sections to provide an overview of the retina. Unfortunately, we still don't think that Dgk is a major contributor to the miR-210 Δ phenotype.

- Fig. 5 is missing.

We are not sure to what the reviewer is referring with this comment, as Fig. 5 was included in the manuscript.

Reviewer #3 (Comments to the Authors (Required)):

Weigelt et al. study miRNA expression in four tissue types of *Drosophila melanogaster*. The authors categorize these hundreds of miRNA molecules according to their specific expression patterns and focus on the brain specific miR-210. This miRNA molecule has previously been associated with hypoxia. However, the authors clearly demonstrate that miR-210 regulation doesn't impact hypoxia in *Drosophila*. In contrast, miR-210 deletion results in degeneration of adult photoreceptors linking regulatory functions to the maintenance of these sensory neurons. This novel miR-210-mediated phenotype is well and plausible described and supported by the data. In addition, the authors aim at providing mechanistic insights, however, that aspect appears rather premature. Some claims are also not well supported by data or cited literature.

Major comments:

- Results about miR-958 and miR-314: The authors only present tissue-specific expression patterns. How is this a functional proof for "gut-specific responses to bacterial infection" or "that miR-314 indeed plays an important function in the gut"? These statements are not supported by the data shown.

We didn't state that our data represent a functional proof for the function of miR-958 or miR-314, but we stated that "Our results suggest that the gut-specific miR-958 might contribute to the gut-specific responses to bacterial infection", which is a hypothesis based on the suggested function of this miRNA and our expression data.

For miR-314 a function in the gut has already been demonstrated by Chandra et al. 2015. We consider the tissue-specific expression atlas for the four main adult tissues of the fly a valuable resource for the miRNA community and we highlighted a few miRNAs with already known function.

- How "highly expressed" is miR-210? Is this based on small RNA-Seq, which is known to have high amplification biases? How high is miR-210 expression in relation to other brain-enriched miRNAs such as miR-124?

For the well-known brain-enriched miR-124 we detected around 100 reads (normalized) in each replicate of our miRNA sequencing experiment, but for miR-210-3p between 400 and 500 reads (normalized). We agree that miRNA seq is not the ideal experiment to determine the expression level of miRNAs due to amplification biases, miRNA Northern blotting might help in the future to determine this. However, this is not the scope of our study, as the main finding of our study is the functional analysis of miR-210 in the brain.

- The miR-210 overexpression data during development is interesting and reproduces partly previous work (Cusumano et al. 2018). This could be mentioned in the text. Do the authors also see more than seven rhabdomeres in wildtype eyes when overexpressed?

We do not agree with the reviewer that the overexpression of miR-210 in the eye using the GMR-Gal4 driver reproduces previous work. In their study Cusumano et al, used different clock-cell-specific Gal4 lines and analysed phenotypes related to activity and sleep, but not the eye-specific GMR-Gal4 driver line. While we cite the corresponding paper in a different context, we do not think that the sleep phenotypes need to be included here, also because it is not clear whether the clock-neurons are a site of endogenous miR-210 expression and whether the observed sleep defects present a physiological meaningful phenotype.

As the reviewer suggested we added TEM pictures of miR-210 over-expression mutants in the wildtype background to Figure S3C. Similar to our miR-210 over-

expression rescue experiments, we detected few ommatidia with 8 rhabdomers, which might be split rhabdomers.

- Fig.5 A, B just reflect the phenotype at the transcriptional level, i.e. photoreceptor loss, but do not directly link miR-210 to it. The fact that the highest upregulated gene, Dgk, is a potential miR-210 target but cannot functionally linked to the phenotype requires a deeper discussion. Either the data is wrong, or Targetscan not accurate, or both.

We were also very surprised (and slightly disappointed) that we could not confirm Dgk as functional relevant target of miR-210 given its strong upregulation in miR-210 mutant flies. However, we show by RNAi-mediated knock-down and overexpression that Dgk is neither necessary nor sufficient for the miR-210 dependent eye-phenotype. Furthermore, we controlled knock-down and overexpression efficiency by q-RT-PCR and we now provide evidence that we are able to at least partially rescue the retina degeneration phenotype by using our genetic set up (see below). We think that our experiments are well controlled and therefore concluded that the function of Dgk is not essential in mediating miR-210 dependent retina degeneration. Our results do not exclude Dgk as a direct miR-210 target, and therefore we cannot comment on the accuracy of Targetscan, however, we did not follow this up further given the lack of a clear functional link between Dgk and the miR-210 phenotype.

- In Figure 5C, the authors use two target prediction tools and identify four transcripts with annotated binding sites and upregulation in miR-210 deleted samples. Why do the authors not test and validate these genes (upregulation in wt or knock-down in the miR-210 mutant)?

As suggested by the reviewer we addressed the in vivo function of Apc, Fasn1 and Vha55 by knocking them down in the miR-210 Δ mutant background. While we did not detect a rescue upon knock-down of Apc and Vha55, interestingly, we found that knock-down of Fasn1 partially rescued the ERG defects of miR-210 Δ . We followed this up further, by overexpression of Fasn1 in wild type flies, however, this did not lead to retina degeneration, suggesting that Fasn1 is necessary but not sufficient to cause miR-210 dependent retina degeneration. Whether Fasn1 is really an in vivo

target of miR-210 is currently unclear, as we were not able to demonstrate this directly by using an in vitro luciferase set up. Noteworthy, we were able to rescue the eye phenotype of miR-210 mutants by overexpression of the triacylglyceride lipase Brummer, demonstrating that both decreased lipogenesis and increased lipolysis can rescue miR-210 dependent eye degeneration. We added these new data to the manuscript as Fig 5D-F.

- Does evolutionary conservation of a miRNA sequence correlate with their function in different species and tissues? This implication is not well justified. The miR-210 phenotype in flies shows characteristics of the miR-182 and miR-183 phenotype in mouse photoreceptors (Buskamp et al. 2014), suggesting that specific functions are mediated by different miRNAs in different species.

There is evidence that miRNA function can be evolutionarily conserved across long evolutionary distances, e.g. miR-1 plays a role in muscle cells in worms, flies and mice. While we currently do not have evidence that the function of miR-210 in the eye is evolutionarily conserved, the finding that miR-210 is also expressed in the mouse eye warrants further investigations into its function in the mammalian eye. Only because miR-182 and miR-183 are important for eye function in mice does not exclude a role for miR-210 in this process. We have now highlighted the role of the miR-183/96/182 cluster by adding a sentence "In mice, other eye-specific miRNAs such as the miR-183/96/182 cluster have been shown to play a crucial role in photoreceptor maintenance (Lumayag et al. 2013, Buskamp et al. 2014), demonstrating that miRNAs indeed are essential for vision in mammals."

- The claim that miR-210 is "one of the highest expressed miRNAs in the mouse retina" is also not well justified. One cannot directly extract expression levels from small RNA-Seq experiments. Krol et al. only mention miR-210 in a supplementary table.

We think that small RNA sequencing is at least a good indication, but we agree that further studies (e.g. Northern blotting) might be helpful in the future to prove this point.

We adjusted the sentence to: “Furthermore, previous studies have demonstrated that miR-210 is *highly* expressed in the mouse retina (...)”

- Linking miR-210 function to human phenotypes has also to be taken with extreme care as miR-210 was not a hit in a human retina study (Karali et al. 2016). Therefore, projecting conserved photoreceptor maintenance functions of miR-210 from flies to mammals is pure speculation as hypoxia via miR-210 was not conserved. The implied "conserved" impact of miR-210 is not plausible.

There is abundant evidence that miR-210 is expressed in the mouse eye (Xu *et al.* 2007, Hackler *et al.* 2010, Karali *et al.* 2016) including the mouse retina, and while miR-210 was not identified in the human eye by deep sequencing in one study (Karali et al. 2016), miR-210 expression has been detected in the eye in another study (Ragusa et al., 2013). In addition there is evidence for an association of a miR-210 binding site in human patients with age-related macular degeneration (Ghanbari et al. 2017). We have added this information now to the discussion. Whether the function of miR-210 is conserved from flies to mice and maybe even humans is currently unclear, but we feel that our works provides the motivation to study miR-210 function in the mammalian eye.

Minor comments:

- Please show sample numbers 'n' in figures or legend and text and also provide p-values for statistical significance throughout the text.

We added this information in the figure legend and where appropriate in the text.

- Fig.1B: Why is the miR-210 expression only significantly different between brain and fat body?

miR-210 is also significantly different expressed compared to the other tissues tested. We did not include this information previously to keep the figure simple but have included it now.

- Fig.3A, D: For underlying a significant difference between wt and miR-210 knockout, a quantification of the phenomena shown in the images is needed.

We agree with the reviewer that quantification is essential to interpret scientific data. However, retinal degeneration in miR-210 Δ mutants is very severe at later stages and therefore the unambiguous identification of individual ommatidia or photoreceptor cells is not always possible, which makes quantification difficult. To overcome this problem we used electroretinography, which allows solid quantification. We feel that the combination of ERG and the images provides a solid basis for our conclusions.

January 7, 2019

RE: Life Science Alliance Manuscript #LSA-2018-00149-TR

Dr. Sebastian Grönke
Max-Planck Institute for Biology of Ageing
Biological Mechanisms of Ageing
Joseph-Stelzmann Str. 9b
Cologne, NRW 50931
Germany

Dear Dr. Grönke,

Thank you for submitting your revised manuscript entitled "Loss of miR-210 leads to progressive retinal degeneration in *Drosophila melanogaster*". As you will see, the reviewers appreciate the introduced changes and reviewer #2 provides constructive guidance for further improvements of the manuscript by minor text changes. We would thus be happy to publish your paper in Life Science Alliance pending these final revisions as well as revisions necessary to meet our formatting guidelines:

- please note that you mention a Figure 4D in the text => please correct (to 3D)
- please note that figure panel S1E is currently not called out => please add the callout in the manuscript text.

A. FINAL FILES:

-- High-resolution figure, supplementary figure and video files uploaded as individual files: See our detailed guidelines for preparing your production-ready images, <http://life-science-alliance.org/authorguide>

B. MANUSCRIPT ORGANIZATION AND FORMATTING:

Full guidelines are available on our Instructions for Authors page, <http://life-science-alliance.org/authorguide>

Sincerely,

Reviewer #1 (Comments to the Authors (Required)):

In the revised version of the manuscript "Loss of miR-210 leads to progressive retinal degeneration in *Drosophila melanogaster*" essentially almost all my concerns are addressed - either by new

experiments or by changes to the text. Authors did not test via qPCR the expression levels of the four targets which were predicted by both databases (Apc, CG5554, Fasn 1, Vha55). However, I think that authors did a good job in answering critiques and changing figures to make the manuscript better. In my view, the manuscript is now suitable for publication in Life Science Alliance.

Reviewer #2 (Comments to the Authors (Required)):

The authors have addressed nearly all points raised in my previous review and have modified/supplemented the text accordingly, thereby improving its quality.

There are a few points that I would like to bring to the attention of the authors, and they may consider them.

1. They added the following sentence into the "Maintenance of flies" chapter:

"For miRNA sequencing, we dissected brains, thorax (thorax without the gut), fat body (abdomen without the gut and ovaries) and the gut (midgut without malpighian tubules and without the crop)."

I suggest to put this sentence either into the paragraph "RNA extraction and cDNA synthesis" or into "RNA sequencing of miRNAs"

2. Fig. 2A: what is the evidence that the staining on the top are the ocelli?

3. In the chapter "Loss of miR-210 leads to retinal degeneration" (page 5) they write (line 6/7): "... miR-210 is expressed in photoreceptor cells in the lamina and medulla ...". Photoreceptors are not IN the lamina or the medulla, they project into the lamina or medulla.

4. Page 6, line 10: the holes in the retina should not be called vesicles, but rather vacuoles as done later in the text. The authors may consider to use the term "lacunae", as introduced by Ferreiro et al., 2018 (PMID: 29354028). In this paper, the authors describe the lacunae as part of the white mutant phenotype. Question: have the authors checked whether the lacunae observed in miR-210 mutants are due to a white mutation in the background? This could explain the rescue phenotype described in Fig. 4C, since the UAS- and the GAL4 constructs carry w[+], I guess.

5. Page 7: the authors write that retinal degeneration is "independent of light and apoptosis". Which Gal4 line was used to express p35? It would be interesting to read what the authors think about possible mechanism of degeneration, if not p35-mediated apoptosis. In addition, they write that overexpression of p35 did not rescue the receptor potential of miR-210 mutants (Fig. S4D). Did they score the histology? Perhaps degeneration is rescued, but receptor potential not.

6. Recommendation to the authors for future manuscripts: it would have helped the reviewer if pages, figures and lines were numbered.

Dear Dr. Leibfried,

We are very happy to hear that you accepted our manuscript for publication in Life Science Alliance. We have addressed the additional comments made by you and reviewer 2 and also formatted the manuscript following your guidelines. Please find our response below.

- please note that you mention a Figure 4D in the text => please correct (to 3D)
Thank you for pointing out this mistake, we have corrected it in the text.

- please note that figure panel S1E is currently not called out => please add the callout in the manuscript text.
We have added the callout in the text.

Reviewer #2 (Comments to the Authors (Required)):

The authors have addressed nearly all points raised in my previous review and have modified/supplemented the text accordingly, thereby improving its quality.

There are a few points that I would like to bring to the attention of the authors, and they may consider them.

1. They added the following sentence into the "Maintenance of flies" chapter:
"For miRNA sequencing, we dissected brains, thorax (thorax without the gut), fat body (abdomen without the gut and ovaries) and the gut (midgut without malpighian tubules and without the crop)."
I suggest to put this sentence either into the paragraph "RNA extraction and cDNA synthesis" or into "RNA sequencing of miRNAs"

As suggested by the reviewer we have moved the sentences to the paragraph "RNA sequencing of miRNAs".

2. Fig. 2A: what is the evidence that the staining on the top are the ocelli?

We concluded that the observed pattern must be the ocelli, as the position and shape within the tissue was similar to previously published ocelli stainings (e.g. Bernando-Garcia et al., 2016, Fly)

3. In the chapter "Loss of miR-210 leads to retinal degeneration" (page 5) they write (line 6/7): "... miR-210 is expressed in photoreceptor cells in the lamina and medulla ...".
Photoreceptors are not IN the lamina or the medulla, they project into the lamina or medulla.

We thank the reviewer for pointing out this mistake and we have changed the sentence into "... miR-210 is expressed in photoreceptor cells projecting into the lamina and medulla ...".

4. Page 6, line 10: the holes in the retina should not be called vesicles, but rather vacuoles as done later in the text. The authors may consider to use the term "lacunae", as introduced by Ferreira et al., 2018 (PMID: 29354028). In this paper, the authors describe the lacunae as part of the white mutant phenotype. Question: have the authors checked whether the lacunae observed in miR-210 mutants are due to a white mutation in the background? This could

explain the rescue phenotype described in Fig. 4C, since the UAS- and the GAL4 constructs carry w[+], I guess.

As suggested, we have changed vesicles into vacuoles and we agree with the reviewer that this is the better term to use in this context. Concerning the “lacunae” phenotype observed in miR-210 Δ mutant flies we think it is unlikely that this is caused by the white mutation in the background based on the following reasons. We did not observe any “lacunae”/vacuoles in our wDah white-eyed wildtype control flies (Fig 3D). Furthermore, As stated in the Material & Methods part, we always matched the eye colour of our mutants to the eye colour of the control flies, as eye colour is known to affect eye function including the shape and size of the ERG. This also applies to the rescue experiment in Fig. 4C, in which all genotypes carry the mini white marker genes within the UAS or GMR-Gal4 constructs, respectively.

5. Page 7: the authors write that retinal degeneration is "independent of light and apoptosis". Which Gal4 line was used to express p35? It would be interesting to read what the authors think about possible mechanism of degeneration, if not p35-mediated apoptosis. In addition, they write that overexpression of p35 did not rescue the receptor potential of miR-210 mutants (Fig. S4D). Did they score the histology? Perhaps degeneration is rescued, but receptor potential not.

We did not use a Gal4 line, but a construct expressing p35 under the control of the GMR promoter (Bloomington #5774). Unfortunately, we did not identify the alternative mechanism that causes the retinal degeneration in our miR-210 Δ mutants. In our hands, ERGs matched very nicely with the histology and is preferred as it allows solid quantification.

6. Recommendation to the authors for future manuscripts: it would have helped the reviewer if pages, figures and lines were numbered.

We apologize for the inconvenience and will keep this in mind for future manuscripts.

January 11, 2019

RE: Life Science Alliance Manuscript #LSA-2018-00149-TRR

Dr. Sebastian Grönke
Max-Planck Institute for Biology of Ageing
Biological Mechanisms of Ageing
Joseph-Stelzmann Str. 9b
Cologne, NRW 50931
Germany

Dear Dr. Grönke,

Thank you for submitting your Research Article entitled "Loss of miR-210 leads to progressive retinal degeneration in *Drosophila melanogaster*". We appreciate the introduced changes and it is a pleasure to let you know that your manuscript is now accepted for publication in Life Science Alliance. Congratulations on this interesting work.

DISTRIBUTION OF MATERIALS:

Again, congratulations on a very nice paper. I hope you found the review process to be constructive and are pleased with how the manuscript was handled editorially. We look forward to future exciting submissions from your lab.

Sincerely,

Andrea Leibfried, PhD
Executive Editor
Life Science Alliance
Meyershofstr. 1
69117 Heidelberg, Germany
t +49 6221 8891 502
e a.leibfried@life-science-alliance.org
www.life-science-alliance.org